# Gq neuromodulation of BLA parvalbumin interneurons induces burst firing and mediates fear-associated network and behavioral state transition in mice

Xin Fu[1,2,6], Eric Teboul[3,6], Grant L. Weiss[3], Pantelis Antonoudiou[3], Chandrashekhar D. Borkar[2,4], Jonathan P. Fadok [2,4], Jamie Maguire [3 ✉] & Jeffrey G. Tasker [2,5 ✉]

Patterned coordination of network activity in the basolateral amygdala (BLA) is important for fear expression. Neuromodulatory systems play an essential role in regulating changes between behavioral states, however the mechanisms underlying this neuromodulatory control of transitions between brain and behavioral states remain largely unknown. We show that chemogenetic Gq activation and α1 adrenoreceptor activation in mouse BLA parvalbumin (PV) interneurons induces a previously undescribed, stereotyped phasic bursting in PV neurons and time-locked synchronized bursts of inhibitory postsynaptic currents and phasic firing in BLA principal neurons. This Gq-coupled receptor activation in PV neurons suppresses gamma oscillations in vivo and in an ex vivo slice model, and facilitates fear memory recall, which is consistent with BLA gamma suppression during conditioned fear expression. Thus, here we identify a neuromodulatory mechanism in PV inhibitory interneurons of the BLA which regulates BLA network oscillations and fear memory recall.

[1] Neuroscience Program, Tulane University, New Orleans, LA 70118, USA. [2] Tulane Brain Institute, Tulane University, New Orleans, LA 70118, USA. [3] Department of Neuroscience, Tufts University School of Medicine and Graduate School of Biomedical Sciences, Boston, MA 02111, USA. [4] Department of Psychology, Tulane University, New Orleans, LA 70118, USA. [5] Department of Cell and Molecular Biology, Tulane University, New Orleans, LA 70118, USA. [6] These authors contributed equally: Xin Fu, Eric Teboul. ✉email: jamie.maguire@tufts.edu; tasker@tulane.edu

S witching between different brain and behavioral states is necessary to adapt to an ever-changing environment. Accompanying brain-state transitions are prominent changes in population-level rhythmic and synchronous neural activity, which can be detected by changes in oscillations of local field potentials and are shaped by the activity of inhibitory interneurons[1,2]. Subcortical neuromodulatory systems influencing various cognitive processes, such as arousal, are key mediators of brain-state transitions[3,4], and effectively modulate network rhythmic and synchronous patterning. Increasing evidence indicates that inhibitory interneurons are major targets of neuromodulation and are activated by multiple neuromodulators, such as norepinephrine, serotonin, and acetylcholine, via their cognate G protein-coupled receptors (GPCRs) coupled to Gq signaling pathways[5,6]. Thus, Gq-mediated neuromodulatory regulation of inhibitory interneurons may be critical for orchestrating the changes in neural oscillations that underlie operational state transitions of the brain.

The basolateral amygdala (BLA) acts as a critical node among limbic networks for processing emotionally salient information and patterned BLA network activity is central to this process[7–13]. Particular importance has been given to BLA theta- and gamma-frequency activity (2–12 Hz, 40–120 Hz, respectively) in the dynamic regulation of conditioned fear expression[7,13]. Mechanisms of gamma rhythm generation in other limbic regions point to a coordinating role of parvalbumin-expressing (PV) interneurons[1,14–17], and PV interneurons in the BLA have been shown to play a critical role in orchestrating network and behavioral states[10,11,18], including the behavioral expression of fear[10,11]. Parvalbumin-expressing interneurons in cortical areas are sensitive to neuromodulatory signals that tune their activity via GPCR activation to execute brain state-dependent behavioral tasks[19,20]. Thus, Gq signaling in PV interneurons in the BLA may provide a cellular mechanism for the modulatory control of brain and behavioral states. For example, neuromodulatory signaling in the BLA by norepinephrine (NE) plays a well-documented role in promoting conditioned fear states[21]. However, how neuromodulation regulates PV interneuronal circuits and how this influences BLA network rhythms associated with fear states is not known.

Here, we tested the role of Gq neuromodulation of BLA PV interneurons in controlling network states and regulating conditioned fear expression. Our findings reveal molecular and cellular mechanisms for emotional brain-state transitions whereby Gq signaling reconfigures BLA PV interneuron activity patterns, which tunes BLA network oscillations and facilitates conditioned fear expression. Stimulation of Gq signaling in BLA PV interneurons by the activation of exogenous Gq-coupled designer receptors (hM3D) or endogenous α1A adrenoreceptors generated phasic firing in the PV interneurons, which resulted in synchronized phasic bursts of IPSCs and phasic firing in BLA principal neurons. Gq activation via hM3D or α1A adrenergic receptors in BLA PV interneurons suppressed BLA gamma oscillations in ex vivo slices and in vivo. Selective rescue of α1A adrenergic signaling or activation of hM3D receptors in BLA PV interneurons in global α1A adrenoreceptor knockout mice enhanced conditioned fear expression. These findings reveal a role for Gq neuromodulation of BLA PV interneurons in the coordination of BLA network states underlying fear memory expression.

## Results

**Chemogenetic Gq activation in BLA PV interneurons stimulates repetitive bursts of IPSCs.** To investigate the function of Gq activation in PV interneurons in modulating BLA neural activity, a Cre-dependent AAV virus (AAVdj-DIO-hDLX-

hM3D(Gq)-mCherry) was injected bilaterally into the BLA of PV-Cre mice to express Gq-coupled designer receptors exclusively activated by designer drugs (DREADDs) specifically in PV interneurons (Fig. 1a, b). Two weeks after virus injections, whole-cell voltage clamp recordings were performed in putative BLA principal cells in amygdala slices in the presence of glutamate AMPA-receptor (DNQX, 20 µM) and NMDA-receptor antagonists (APV, 40 µM) to isolate inhibitory postsynaptic currents (IPSCs) (Fig. 1c). Notably, Gq-DREADD activation selectively in PV interneurons with clozapine N-oxide (CNO) (5 µM) induced stereotyped phasic bursts of IPSCs in BLA principal neurons. The repetitive IPSC bursts showed an accelerating intra-burst IPSC frequency, which peaked at >50 Hz and generated a shift in the baseline holding current due to summation (Fig. 1d, e). We observed multiple variants of phasic bursts that had varying amplitudes, durations, and acceleration rates (Fig. 1d), suggesting that the different repetitive bursts were generated by different presynaptic PV interneurons. The PV-mediated repetitive IPSC bursts were of relatively long duration (3.63 ± 0.26 s) and recurred at a low frequency (burst frequency: 0.032 ± 0.0015 Hz; inter-burst interval range: 15 to 80 s) (Fig. 1f), and continued for >20 min after CNO was washed from the recording chamber. Consistent with the P/Q-type calcium channel dependence of synaptic output from PV interneurons, and not N-type calcium channel dependence[22–24], we found that the CNO-induced IPSC bursts in BLA principal neurons were abolished by the selective P/Q-type calcium channel blocker ω-agatoxin (0.5 µM), but not by the N-type calcium channel blocker ω-conotoxin (1 µM) (Fig. 1h; Supplementary Fig. 1a–d).

To confirm that the Gq-DREADD-induced IPSC bursts were mediated by the excitation of PV interneurons through Gq signaling, we blocked Gq protein activation with a selective Gα$_{q/11}$ inhibitor YM-254890 (10 µM), which blocks the switch of Gα$_q$ from the GDP- to GTP-bound state[25]. We found that the CNO-induced IPSC bursts were eliminated with the Gq blocker (Fig. 1g, h). In addition, blocking spiking activity with tetrodotoxin (TTX, 0.5 µM) also abolished the PV-Gq-mediated IPSC bursts (Fig. 1g, h), demonstrating the dependence of Gq-induced IPSC bursting in BLA principal neurons on Gq activation in presynaptic PV interneuron somata/dendrites. To determine whether bath application of CNO, which creates a stable drug concentration over minutes, generates a continuous depolarization of PV interneurons that is converted to a phasic synaptic output, we tested whether sustained PV neuron excitation independent of Gq activation is sufficient to generate the repetitive IPSC bursting pattern in principal cells using photoactivation of PV neurons with channelrhodopsin (ChR2). Two weeks after delivery of AAV9-EF1a-DIO-ChR2-mCherry to the BLA of PV-Cre mice, continuous photostimulation of PV interneurons with blue light in brain slices failed to generate phasic IPSC bursts in the principal cells, but induced a tonic increase in IPSCs that was phase-locked to the light stimulation (n = 11 cells from two mice, Supplementary Fig. 1e, f). These results together suggest that Gq signaling in PV interneurons is required for the generation of the phasic pattern of inhibitory synaptic output.

The repetitive bursting pattern of IPSCs in BLA principal neurons suggests that presynaptic PV interneurons fire phasic action potential bursts with Gq activation. To directly test this, extracellular loose-seal patch clamp recordings of PV interneurons were performed in slices from PV-Cre mice expressing Cre-dependent Gq-DREADDs in the BLA (Fig. 1i). Like the pattern of IPSC bursts in principal neurons, PV cells responded to CNO (5 µM) by firing repetitive accelerating bursts of action potentials that recurred at a low frequency (0.034 ± 0.005 Hz, n = 9 cells from 5 mice) in the presence of DNQX and APV (Fig. 1j–m). Further blockade of fast GABAergic inhibitory synaptic transmission with picrotoxin (50 µM) did not affect the pattern of action potential

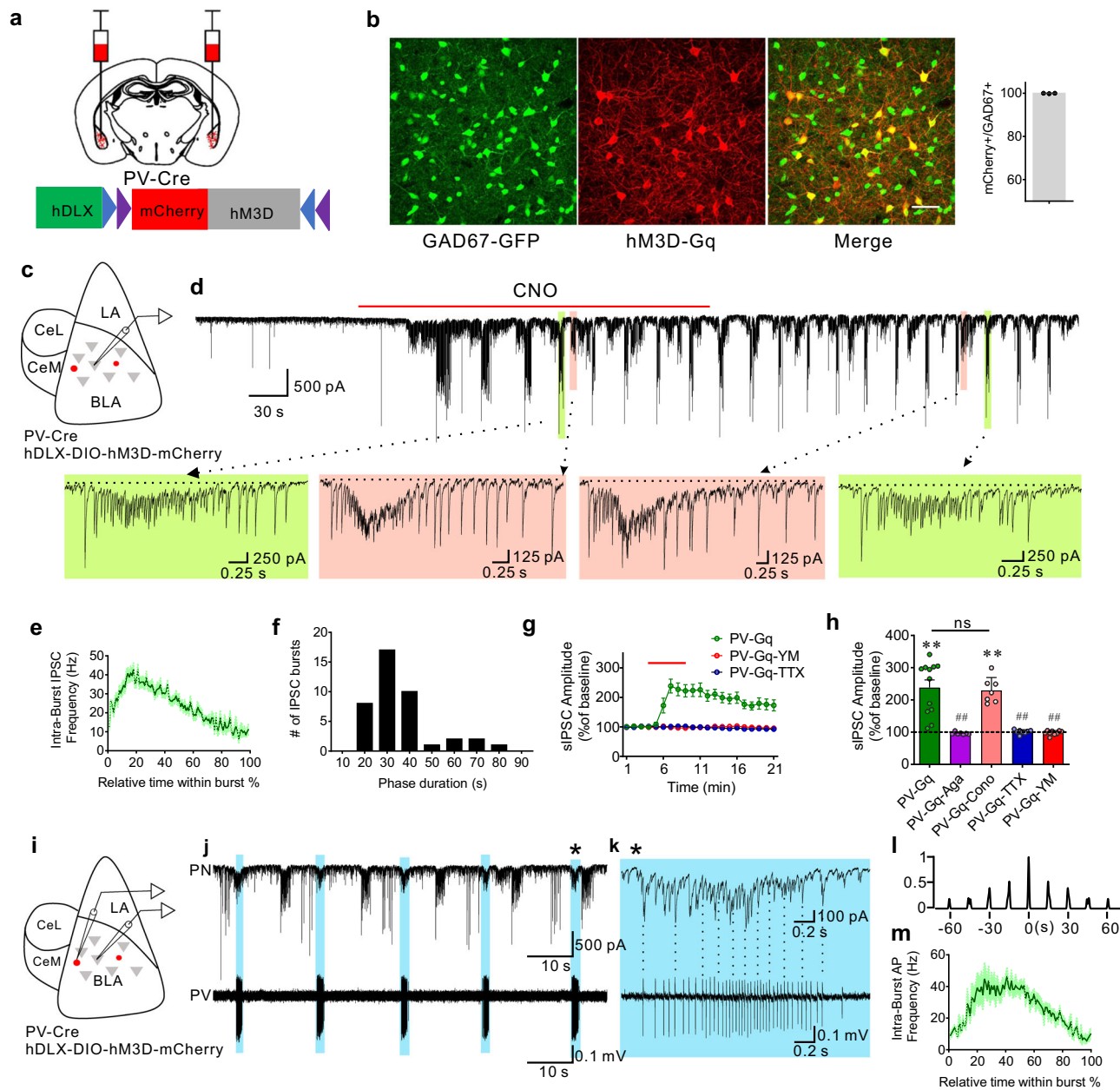

bursts stimulated by CNO in PV interneurons ($n = 5$ cells from 2 mice, Supplementary Fig. 1g), suggesting that the bursting activity in PV cells is mediated by an intrinsic rather than circuit mechanism. In paired recordings of synaptically connected PV and principal neurons ($n = 5$ pairs from 3 mice), we found that the action potential bursts in PV neurons were time-locked to one of the subgroups of IPSC bursts in the principal neurons (Fig.1j) in all 5 paired recordings, confirming that the different variant groups of Gq-induced IPSC bursts in principal neurons were generated by different presynaptic PV interneurons. Moreover, we observed that the action potentials within the bursts of PV cells were time-locked with individual IPSCs in the corresponding IPSC bursts of the principal cells (Fig. 1k). Therefore, Gq activation drives a repetitive bursting pattern of action potentials in BLA PV interneurons to induce patterned bursts of phasic inhibitory synaptic inputs to BLA principal neurons.

Perisomatic PV interneurons innervate hundreds of principal neurons to effectively control spike timing and population-level neural activity[22,26–28], raising the question whether PV-mediated

phasic IPSC bursts are synchronized between BLA principal neurons. To test this, we first examined the synchronization of Gq-induced IPSC bursts with paired recordings from adjacent BLA principal neurons (inter-neuron distance ≤ 40 µm) following CNO activation of hM3D in PV interneurons (Fig. 2a). Interestingly, CNO application induced similar responses in both cells in the majority of the paired recordings (Fig. 2b). Moreover, we found that the CNO-induced recurrent IPSC bursts were synchronized between each pair at two levels: (1) the bursts that were synchronized between two cells were synchronous at each repetition, and (2) the individual IPSCs within synchronized bursts were synchronized (Fig. 2b). This is consistent with each group of repetitive IPSC bursts being generated by phasic action potential bursts in a distinct presynaptic PV interneuron. Dividing the total number of PV-mediated IPSC bursts by the number of synchronized bursts, we calculated that one BLA principal neuron receives, on average, bursting IPSC inputs from 4.1 presumably different presynaptic PV interneurons (Fig. 2c), and that 67.7%, or two thirds, of all the IPSC bursts are synchronized between the pairs of recorded cells (16 recorded pairs from 6 mice).

**Fig. 1 Gq activation in PV interneurons stimulates patterned IPSC bursts in principal neurons. a** Schematic of injection of conditional AAV expressing Gq-coupled DREADD (hM3D) in BLA of PV-Cre mouse. **b** hM3D-mCherry transduced in PV neurons is localized to GABA neurons of Gad67-GFP knockin mice (ten sections, 278 cells from 3 mice, ratio = 99.81%). Scale bar, 50 μm. **c** Schematic showing whole-cell recordings of BLA principal neurons in Gq-DREADD-injected PV-Cre mice. **d** A representative recording in a BLA principal neuron showing the generation of phasic IPSC bursts by selective Gq activation in PV interneurons with bath application of CNO. Dashed arrows designate expanded traces of individual bursts to illustrate different stereotyped IPSC bursts (color-coded) at two successive time points. The dashed lines in the expanded traces show the depressed baseline of the holding current due to the summation of high-frequency IPSCs in the bursts. **e** Mean (±SEM) of instantaneous intra-burst IPSC frequency over the course of the accelerating IPSC bursts (41 bursts from 12 cells, 5 mice). **f** Distribution of the phase duration of repetitive IPSC bursts; 10-s bins from time −5 s to +5 s. (41 bursts from 12 cells, 5 mice). **g, h** Time course and mean change (±SEM) in sIPSC amplitude with Gq activation in PV interneurons. The CNO-induced increase in sIPSC amplitude was completely blocked by pre-incubation of the slices with the P/Q-type calcium channel antagonist ω-agatoxin (Aga) and the sodium channel blocker TTX, and the selective Gαq/11 inhibitor YM-254890 (YM), but not by the N-type calcium channel antagonist ω-conotoxin (Cono) (PV-Gq, 12 cells from 5 mice; PV-Gq-Agatoxin, 7 cells from 3 mice; PV-Gq-Conotoxin, 7 cells from 3 mice; PV-Gq-TTX, 8 cells from 4 mice; PV-Gq-YM-254890, 9 cells from 4 mice; Paired $t$-tests (two-tailed): PV-Gq vs. baseline, $p = 0.0001$; PV-Gq-Conotoxin vs. baseline, $p = 0.0001$; One-Way ANOVA, $F(4,38) = 22.72$, $p < 0.0001$, Dunnett's multiple comparisons test, PV-Gq vs. PV-Gq-Agatoxin, $p < 0.0001$, PV-Gq vs. PV-Gq-Conotoxin, $p = 0.98$, PV-Gq vs. PV-Gq-TTX, $p < 0.0001$, PV-Gq vs. PV-Gq-YM-254890, $p < 0.0001$; **$p < 0.01$ vs. baseline, ##$p < 0.01$ vs. PV-Gq, ns not significant). **i** Schematic showing simultaneous loose-seal recordings from Gq-DREADD expressing PV interneurons and whole-cell recordings from BLA principal cells. **j, k** Representative paired recordings showing repetitive IPSC bursts recorded in a BLA principal neuron with whole-cell recording and associated action potential bursts recorded in a PV interneuron with loose-seal recording. Correlated activities are labeled with blue shading and the burst marked with an asterisk was expanded to show the time-locked IPSCs and action potentials. Selected synchronous spikes and IPSCs are designated by vertical dotted lines. **l** Autocorrelation diagram showing the rhythmicity of action potential bursts in the PV neuron shown in **j**. **m** Mean instantaneous intra-burst frequency (±SEM) of action potentials in PV interneurons (9 cells from 5 mice). Source data are provided as a Source Data file.

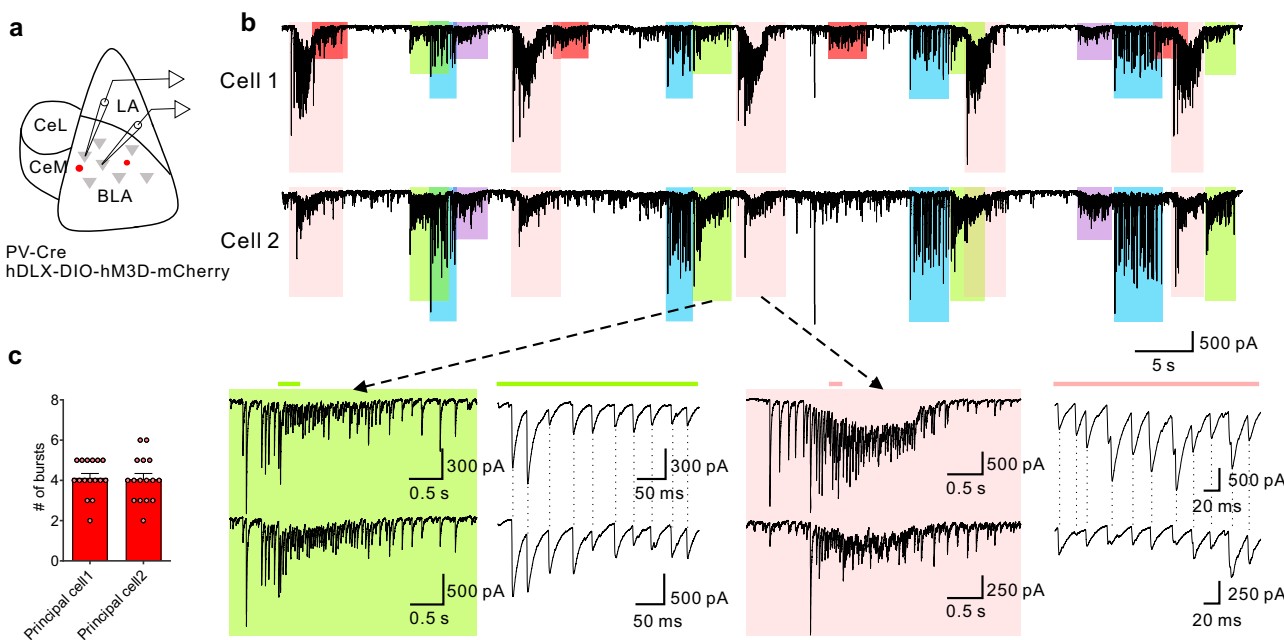

**Fig. 2 Synchronized Gq-activated IPSC bursts in BLA principal neurons. a** Schematic showing paradigm of paired recordings of adjacent BLA principal neurons in Gq-DREADD-injected PV-Cre mice. **b** Representative recordings showing synchronized bursts of IPSCs in a pair of BLA principal neurons, Cell 1 and Cell 2. Different colored boxes represent different IPSC bursts repeated over the course of the recording; the same-colored boxes in the two recordings indicate bursts that were synchronized between the two cells. The red-shaded bursts in cell 1 were not associated with a synchronous IPSC burst in cell 2. Bottom: Expanded traces of two different, color-coded IPSC bursts from each cell. Sections of the bursts were further expanded (indicated by the green and pink lines). The dotted vertical lines show the high synchronicity between the two cells of the individual IPSCs that make up each of the bursts. **c** The total number of different subtypes of bursts (mean ± SEM) induced in pairs of principal cells by Gq activation of PV interneurons (16 principal neuron pairs from 6 mice). Source data are provided as a Source Data file.

**Norepinephrine stimulates PV neuron-mediated repetitive IPSC bursts.** Having established the role for Gq signaling in driving repetitive phasic synaptic outputs from PV interneurons in the BLA, we next tested whether native Gq-coupled GPCRs in PV interneurons generate similar bursts of IPSCs in BLA principal neurons. Norepinephrine (NE) is an arousal neuromodulator that is released in the amygdala during stress[29], and Gq-coupled α1 adrenoreceptors are highly expressed in the BLA[30]. We performed voltage clamp recordings of IPSCs in BLA

principal neurons in the presence of glutamate AMPA- and NMDA-receptor antagonists. NE application (20–100 μM) induced a robust, concentration-dependent increase in spontaneous IPSCs that was characterized by a repetitive bursting pattern, which was like that elicited by Gq-DREADD activation in PV neurons (Fig. 3a). Following an initial increase in IPSC frequency that inactivated within 35–135 s (mean = 71 s) and was not seen with Gq-DREADD activation in PV neurons, a rhythmic bursting pattern of IPSCs emerged at higher NE concentrations

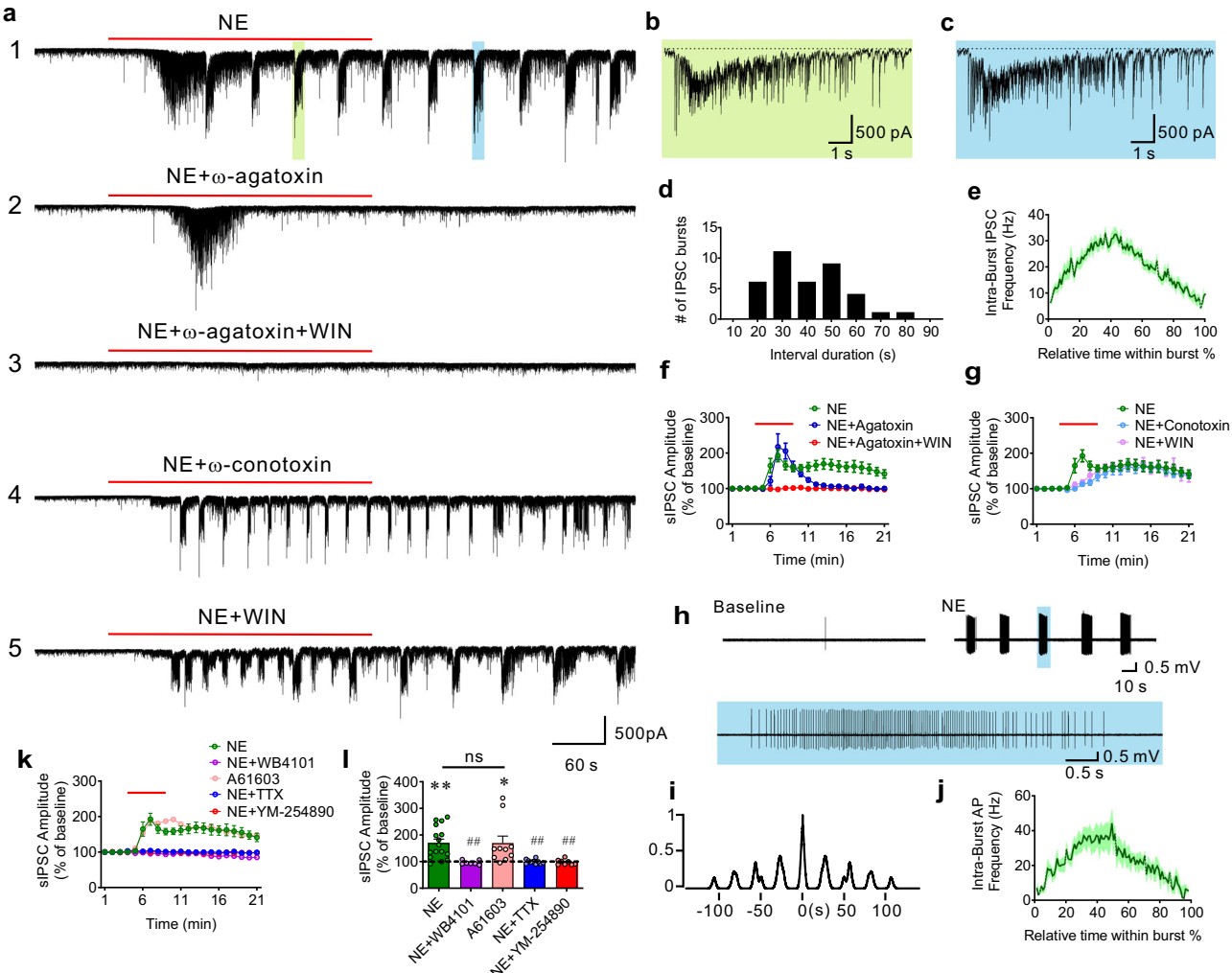

**Fig. 3 α1A adrenergic receptor activation of PV interneurons generates phasic IPSC bursts in BLA principal cells. a1–5** Representative recordings showing the effect of different treatments on NE-induced repetitive IPSC bursts in the BLA principal neurons. **1**. NE application induced phasic IPSC bursts following an initial increase in IPSCs. **2**. Inhibiting PV neuron-mediated transmission with the P/Q calcium channel blocker ω-agatoxin selectively blocked the NE-induced repetitive IPSC bursts, but not the initial increase in IPSCs. **3**. Co-application of ω-agatoxin and the CB1 receptor agonist WIN 55,212-2 blocked both the initial increase in IPSCs and the repetitive IPSC bursts. **4**. Pretreatment of a slice with the N-type calcium channel blocker ω-conotoxin selectively inhibited the initial increase in IPSCs induced by NE, but not the repetitive IPSC bursts. **5**. Application of the CB1 receptor agonist WIN 55,212-2 blocked the NE-induced initial IPSC increase but did not affect the repetitive IPSC bursts. **b**, **c** Expanded traces of individual IPSC bursts in **a1** (indicated with green and blue shading), showing the fast acceleration in the intra-burst IPSC frequency and resulting depression in the baseline holding current (indicated with dashed lines) induced by NE. **d** Histogram showing the distribution of phase durations of NE-induced repetitive IPSC bursts. 10-s bins from time −5 s to +5 s. ($n = 38$ bursts from 16 cells, 5 mice). **e** Mean (±SEM) instantaneous intra-burst IPSC frequency over the course of the NE-stimulated accelerating IPSC bursts ($n = 38$ bursts from 16 cells, 5 mice). **f** Time course of the effect of blocking P/Q-type calcium channels and activating CB1 receptors on sIPSC amplitude. The NE-induced plateau increase in sIPSC amplitude (mean ± SEM) was blocked by ω-agatoxin (P/Q blocker), while the peak increase, corresponding to the NE-induced initial increase in IPSCs shown in **a2**, was unaffected ($n = 10$ cells from 4 mice). The ω-agatoxin-insensitive IPSCs were blocked by CB1 receptor activation with WIN 55,212-2 ($n = 8$ cells from 3 mice), corresponding to the recording in **a3**. **g** Time course of the effect of blocking N-type calcium channels and activating CB1 receptors on sIPSC amplitude (mean ± SEM). Both treatments selectively eliminated the NE-induced initial increase in IPSCs, with little effect on NE-induced repetitive IPSC bursts ($n = 12$ and 10 cells from 4 and 3 mice, respectively), which corresponds to the recordings in **a4** and **a5**. **h** Representative loose-seal extracellular recording showing the NE-stimulated repetitive bursts of action potentials in a PV interneuron. A burst of action potentials indicated by the blue box was expanded below to show the accelerating intra-burst spike frequency. **i** Autocorrelation showing the rhythmicity of NE-induced AP bursts in the PV interneuron shown in **h**. **j** Mean (±SEM) instantaneous intra-burst action potential frequency over the course of PV neuron action potential bursts induced by NE ($n = 7$ cells from 3 mice). **k**, **l** Mean (±SEM) change in sIPSC amplitude over time in response to NE ($n = 16$ cells from 5 mice), NE + α1A receptor antagonist WB4101 ($n = 7$ cells from 4 mice), α1A receptor agonist A61603 ($n = 10$ cells from 4 mice), NE + TTX ($n = 10$ cells from 5 mice), NE + Gq antagonist YM-254890 ($n = 8$ cells from 3 mice); paired $t$-tests (two-tailed): NE vs. baseline, $p = 0.0003$, A61603 vs. baseline, $p = 0.023$; One-Way ANOVA, $F(4, 46) = 6.22$, $p = 0.0004$, Dunnett's multiple comparisons test, NE vs. NE + WB4101, $p = 0.0098$, NE vs. A61603, $p > 0.99$, NE vs. NE + TTX, $p = 0.0052$, NE vs. NE + YM-254890, $P = 0.0078$; *$p < 0.05$, **$p < 0.01$ vs. baseline, ##$p < 0.01$ vs. NE, ns not significant; values at time = 13 min were used to compare ω-agatoxin-sensitive IPSCs. Source data are provided as a Source Data file.

(Fig. 3a–d; Supplementary Fig. 2a, b). The repetitive IPSC bursts occurred at a low frequency ($0.028 \pm 0.0017$ Hz), displayed a fast acceleration of intra-burst IPSC frequency that reached peak frequencies > 50 Hz (peak frequency = 50–133 Hz), lasted for several seconds (duration, 1.58–10.55 s, mean = $4.88 \pm 0.37$ s), and gradually tapered off (Fig. 3b–e); these NE-induced IPSC bursts were very similar to the repetitive IPSC bursts stimulated in principal neurons by Gq activation in PV interneurons (see Fig. 1). In some recordings, we observed multiple variants of NE-induced phasic IPSC bursts with differing burst characteristics (Supplementary Fig. 3a), which we interpreted to be generated by multiple presynaptic PV neurons based on the evidence from the Gq-DREADD-induced IPSC bursts. Individual IPSCs within the NE-induced bursts showed a fast rise time (10-90%: $1.11$ ms $\pm$ $0.07$ ms) and decay time (tau: $18.94$ ms $\pm$ $1.04$ ms), suggesting that they originated from perisomatic inhibitory interneurons, such as cholecystokinin (CCK) or PV basket cells[24,31]. CCK and PV basket cells can be distinguished by differential expression of voltage-gated calcium channels and CB1 receptors at their synapses[22,32]. Double dissociation of NE-induced IPSCs with calcium channel blockers and a cannabinoid receptor agonist showed that while the initial increase in IPSC amplitude was blocked by the N-type calcium channel blocker ω-conotoxin (1 µM) and a CB1 receptor agonist, WIN 55,212-2 (1 µM), the repetitive IPSC bursts were not affected by either ω-conotoxin or WIN 55,212-2, but were selectively blocked by the P/Q-type calcium channel blocker ω-agatoxin (0.5 µM) (Fig. 3a, f, g). Since GABA release from PV interneurons is P/Q-type channel dependent, while GABA release from CCK interneurons is mediated by N-type channels[22,24] and suppressed by CB1 receptor activation[33,34], this indicated that the NE-induced phasic IPSC bursts were generated by GABA release from PV interneuron synapses onto the principal cells. While it abolished the NE-induced IPSC bursts, blocking PV interneuron inputs to principal cells with the P/Q channel antagonist only suppressed the overall increase in IPSC frequency by about 50%, and blocking CCK basket cell inputs with the CB1 receptor agonist failed to reduce the frequency response further (Supplementary Fig. 3b), which suggested that NE also activates other interneuron inputs to the principal cells, possibly from somatostatin cells. The further decrease in the NE-induced increase in IPSC frequency by TTX (Supplementary Fig. 3h–j) suggests that this residual IPSC response to NE is mediated by a combination of NE actions at the somata (TTX-sensitive) and axon terminals (TTX-insensitive) of these unidentified presynaptic interneurons.

To directly test whether NE activates repetitive bursts of action potentials in PV interneurons, extracellular loose-seal recordings of PV interneurons were performed in slices from PV-Cre mice crossed with Ai14 reporter mice, in which PV interneurons express tdTomato. Application of NE (100 µM) induced repetitive bursts of accelerating action potentials in these neurons that recurred at intervals of tens of seconds ($0.03 \pm 0.006$ Hz, $n = 7$ cells from 3 mice), like the NE-induced IPSC bursts in principal neurons (Fig. 3h, j). Note that NE failed to activate PV neurons when they were recorded in the whole-cell current-clamp configuration, suggesting that the intracellular signaling mechanism necessary to generate the NE response was washed out with the dialysis of the cytosol. Overall, these data suggest that activation of PV interneurons by NE generates repetitive IPSC bursts that resemble those stimulated by chemogenetic activation of PV interneurons.

To determine whether the NE-induced IPSC bursts are mediated by activation of Gq-coupled adrenoreceptors, we first tested for the adrenoreceptor subtype dependence of the NE-induced increase in IPSCs. Whereas the β adrenoreceptor antagonist propranolol (10 µM) had no effect on the bursts, the NE-induced IPSC bursts were abolished by the broad-spectrum α1 adrenoreceptor antagonist prazosin (10 µM) (Supplementary Fig. 3d–g). The NE-induced increase in IPSCs was also abolished by the α1A adrenoreceptor-selective antagonist WB4101 (1 µM) and mimicked by the α1A adrenoreceptor-selective agonist A61603 (2 µM) (Fig. 3k, l; Supplementary Fig. 3h–j). These results suggest an α1A adrenoreceptor dependence of the NE-induced inhibitory synaptic inputs to BLA principal cells. Consistent with α1A adrenoreceptor signaling through Gq, blocking Gq activation with the Gα$_{q/11}$ inhibitor YM-254890 eliminated all NE-induced IPSCs (Fig. 3k, l; Supplementary Fig. 3h–j). In addition, blocking spiking activity with TTX also inhibited the NE-induced increase in IPSCs (Fig. 3k, l; Supplementary Fig. 3h–j). Therefore, like the chemogenetically induced phasic IPSC bursts, the NE-induced repetitive IPSC bursts were mediated by Gq activation, likely in presynaptic PV interneurons.

It has been reported that the commercial antibodies against α1 adrenoreceptors are not specific[35]. Therefore, to test whether PV interneurons in the BLA express α1A adrenoreceptors, we took advantage of a global α1A adrenoreceptor knockout mouse line (*Adra1a* KO) in which a lacZ gene cassette is placed in frame with the first exon of the *Adra1a* gene, which allows the visualization of α1A adrenoreceptor expression by histochemical staining for β-galactosidase activity with X-gal[36]. We found the *Adra1a* gene to be expressed at high levels in the cortex, hippocampus, amygdala, and hypothalamus (Fig. 4a). In the BLA, we observed a sparse distribution of X-gal-stained cells, which was suggestive of labeled interneurons. To test for the GABA neuron identity of the X-gal-stained cells, we crossed the *Adra1a* KO mouse with a Gad67-eGFP mouse, in which all inhibitory interneurons in the BLA express GFP[37]. *Adra1a* KO x Gad67-eGFP brain slices were imaged and analyzed for X-gal and GFP co-staining[38]. The sparsely distributed X-gal-labeled cells showed a 98.37% overlap with GFP-labeled GAD67 GABAergic cells (Fig. 4b, c). We next tested for the expression of α1A adrenoreceptors in PV interneurons by injecting a Cre-dependent AAV virus expressing mCherry into the BLA of PV-Cre mice crossed with the *Adra1a* KO mouse (PV-Cre::*Adra1a* KO) and examined X-gal co-labeling of mCherry-positive PV interneurons. Two weeks after virus injection, we observed most of the labeled PV interneurons (84.7 %) to be positive for X-gal (Fig. 4d, e). Together, these data demonstrate that α1A adrenoceptors are selectively expressed in GABAergic interneurons in the BLA, including in most PV interneurons.

Multiple neurotransmitter receptors couple to Gα$_{q/11}$ to enhance neuronal excitability, suggesting other neuromodulators may also induce similar phasic IPSC bursts when acting on PV interneurons. Hence, we also tested the effect of serotonin, another neuromodulator that regulates BLA neural circuits[39] and signals through Gαq/11 on IPSCs in BLA principal neurons. Following the blockade of glutamatergic transmission with DNQX (20 µM) and APV (40 µM) and CCK basket cell-mediated transmission with the CB1 agonist WIN 55,212-2 (1 µM), we found that serotonin (100 µM) also induced repetitive bursts of IPSCs that were inhibited by the P/Q calcium channel blocker, ω-agatoxin (0.5 µM), and by an antagonist of the Gq-coupled 5-HT2A receptor, MDL 100907 (1 µM) (Supplementary Fig. 4). These data suggest that serotonin also stimulates a phasic synaptic output from PV interneurons, and indicate that Gq activation of PV interneurons serves as a general cellular mechanism for different neuromodulators to regulate BLA neural circuit activity under different emotional states.

**Chemogenetic Gq signaling in PV interneurons alters BLA network states**. PV interneurons are critically involved in the generation of gamma-frequency oscillations, which are phase-locked

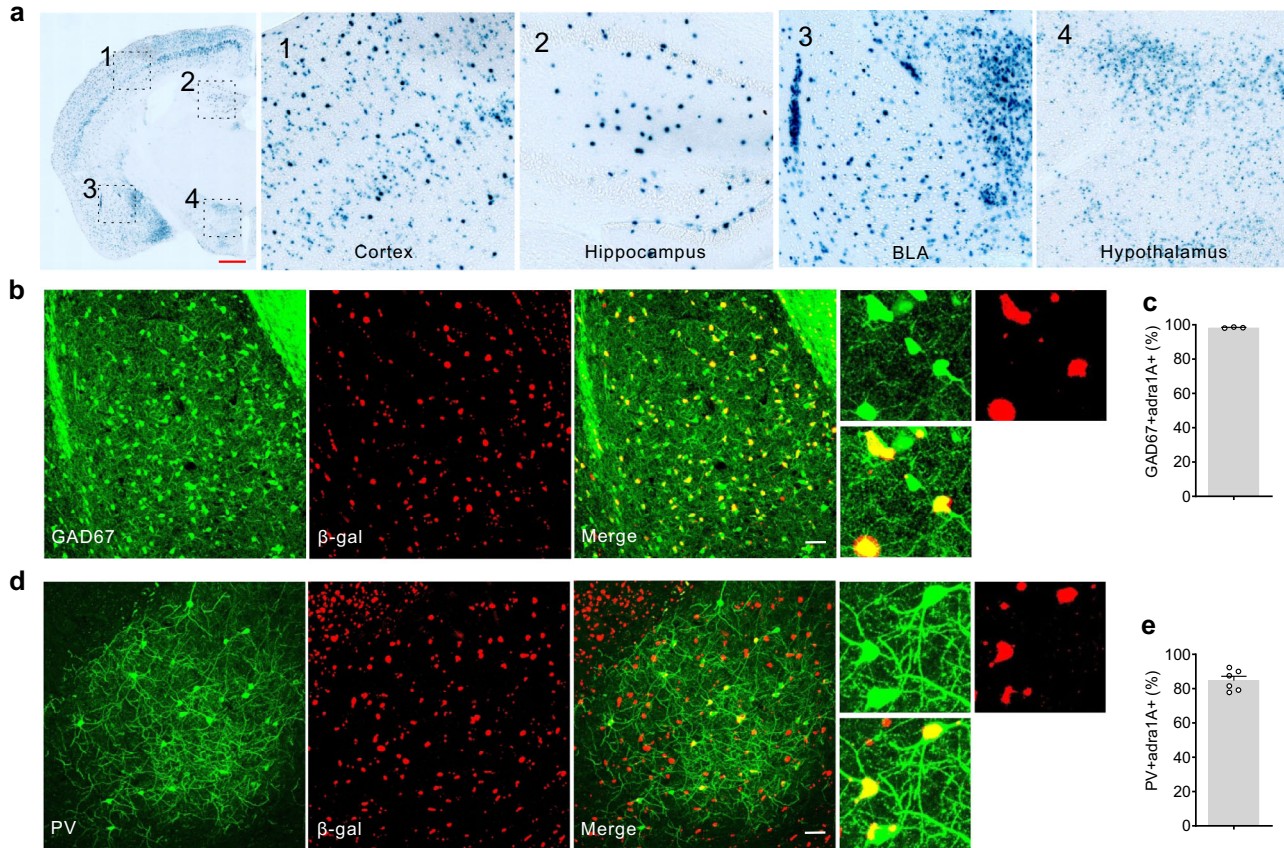

**Fig. 4 Expression of α1A adrenoreceptors in BLA PV interneurons. a** Coronal section of the brain showing β-galactosidase staining of α1A adrenoreceptor-expressing cells in the *Adra1a* KO mouse. 1-4 High-magnification images of areas indicated by dashed boxes showing the cortex (1), the hippocampus (2), the BLA (3), and the hypothalamus (4). Scale bar, 500 μm. **b** Colocalization of the β-gal signal with GFP expression in GABA interneurons in the BLA. Scale bar, 50 μm. **c** Mean (±SEM) percentage of total GAD67-positive cells in the BLA that were positive for β-gal (98.4 ± 0.18%, *n* = 302 cells in three sections from 2 mice). **d** Colocalization of the β-gal signal with mCherry expression in PV interneurons labeled by injection of Cre-dependent virus in PV-Cre mice. Scale bar, 50 μm. **e** Mean (±SEM) percentage of mCherry-labeled PV interneurons that were positive for β-gal (84.7 ± 2.46%, *n* = 134 cells in six sections from 2 mice).

to PV neuron action potentials during tonic high-frequency spiking activity[1,15,16]. As Gq activation in PV interneurons generated phasic action potentials in PV cells and synchronized repetitive IPSC bursts in principal cells of the BLA, we postulated that Gq activation of PV cells may reconfigure pattern generation among BLA networks and alter potential gamma-frequency network coordination. To address this, we first recorded the spiking responses in BLA PV interneurons and principal cells to PV neuron Gq activation during tonic high-frequency spiking activity.

Loose-seal patch clamp recordings were performed to record spiking activity in hM3D-Gq-transduced PV interneurons using an extracellular potassium concentration that was increased from 2.5 mM to 7.5 mM to increase spontaneous spiking. Strikingly, in all the PV interneurons that displayed spontaneous tonic firing under these conditions (*n* = 7 cells from 3 mice, range: 9.7–32.2 Hz, mean: 18.56 ± 3.4 Hz), bath application of CNO (5 μM) transformed the pattern of PV neuron activity from tonic to phasic spiking (Fig. 5a, b). The tonic to phasic transformation is also reflected by a leftward shift in the interspike interval (ISI) and an increase in the predominant firing rate and ISI coefficient of variation after CNO application (Fig. 5c–e)[40]. CNO application did not alter the firing pattern in PV interneurons transfected with mCherry alone (*n* = 6 cells from 3 mice, range: 7.3–18.7 Hz, mean: 11.56 ± 1.9 Hz) (Supplementary Fig. 5a–e). This indicated that Gq activation in PV interneurons causes a switch in the operational mode of BLA PV cells from tonic to phasic activity.

To determine the effect of the PV-mediated phasic IPSC bursts on the firing pattern of BLA principal neurons, we recorded from principal neurons in the whole-cell current-clamp recording configuration with a potassium gluconate solution in the recording pipette and activated Gq-DREADDs in PV interneurons with bath application of CNO (5 μM). The membrane potential of recorded principal cells was held above threshold with positive current injection to elicit tonic 2–5 Hz action potential firing, the range of spontaneous spiking activity observed in these cells in vivo[41,42]. Consistent with the change in baseline holding current of principal neurons caused by the accelerating IPSC bursts, activation of repetitive IPSC bursts with CNO application induced prominent oscillatory hyperpolarizations of the membrane potential, shifting the firing pattern of BLA principal cells from tonic to phasic, characterized by slowly oscillating spike bursts at 0.035 ± 0.006 Hz (Fig. 5f, g, *n* = 7 cells from three mice). Together, these data reveal a regulatory role for fast-spiking interneurons in controlling BLA principal neuron activity patterns in response to Gq activation.

Given the robust influence of Gq signaling on the pattern of BLA PV interneuron activity, we postulated that Gq activation in PV neurons may reconfigure BLA population-level neural activity. To test this, we recorded pharmacologically induced BLA gamma oscillations (30–80 Hz) using a recently developed ex vivo slice model[18]. Local field potentials (LFPs) were recorded in BLA slices with transected hippocampal inputs from mice with

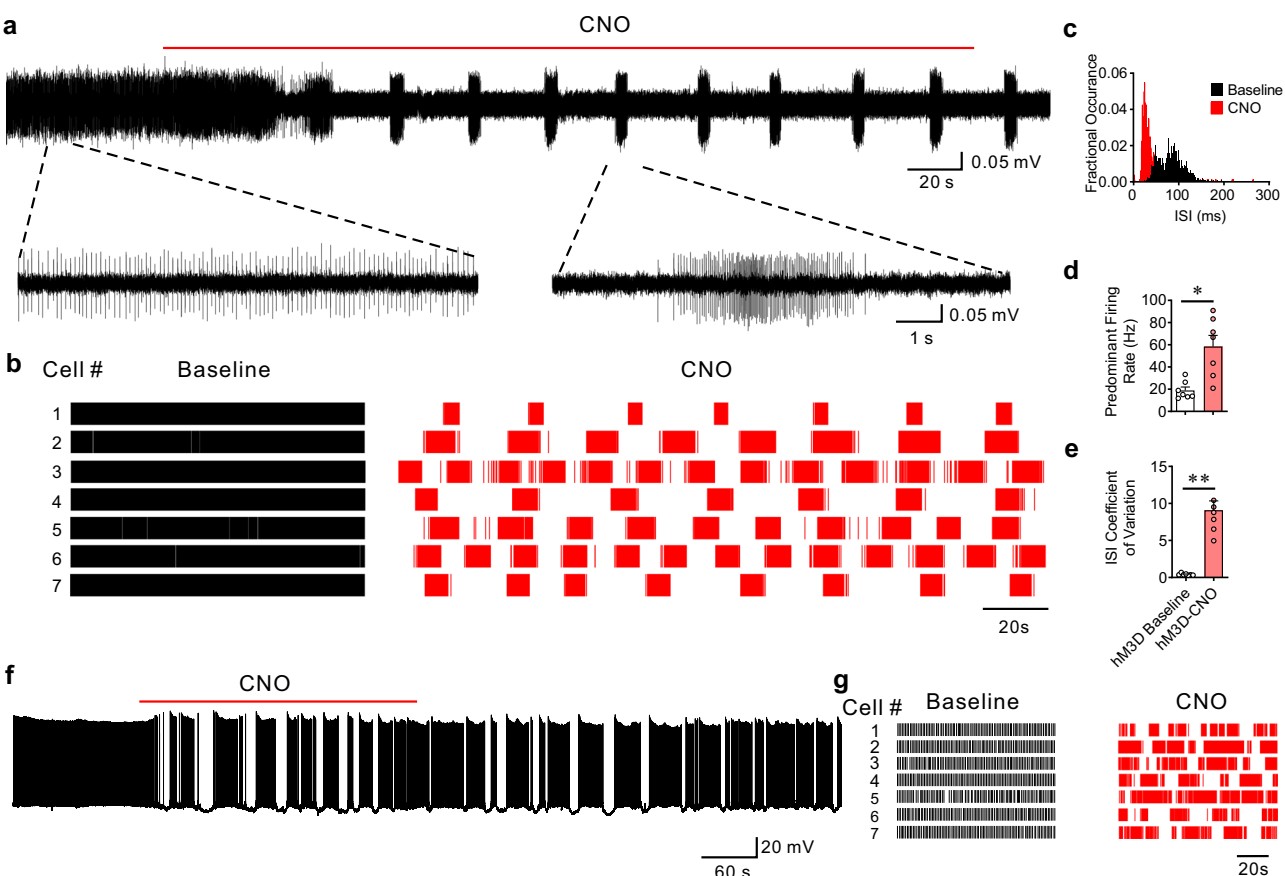

**Fig. 5 Burst firing in PV interneurons transforms BLA neural activity. a** A representative recording showing that chemogenetic activation of PV interneurons switches the firing pattern of PV cells from tonic to phasic. Below: Traces were expanded to show the tonic firing of the baseline and burst firing after CNO application. **b** Raster plots of spiking activity in recordings from 7 PV interneurons (from 3 mice) that show the transformation of tonic spiking to phasic spiking by CNO. **c** Representative interspike interval (ISI) histogram of the cell shown in **a**. **d**, **e** Gq activation increased the predominant firing rate (**d**) and the ISI coefficient of variation (**e**) in the BLA PV interneurons (mean ± SEM, $n = 7$ cells from three mice, paired $t$-tests (two-tailed): predominant firing rate, CNO vs. baseline, $p = 0.013$; ISI coefficient of variation, CNO vs. baseline, $p = 0.0006$, $*p < 0.05$, $**p < 0.01$). **f** A representative current-clamp recording from a principal neuron showing that Gq activation in PV cells transforms the firing pattern of a BLA principal neuron from tonic to phasic. **g** Raster plots of spiking activity in recordings from seven principal neurons (from 3 mice) that show the transformation of tonic spiking to phasic spiking by CNO. Source data are provided as a Source Data file.

hM3D-Gq-expressing PV cells (Supplementary Fig. 6a). Bath application of CNO (5 μM) robustly suppressed gamma oscillatory power (Supplementary Fig. 6b–d). We next examined whether α1A adrenoreceptor activation could similarly modulate BLA gamma oscillations in slices from wild-type mice. Consistent with hM3D-Gq activation, bath application of 100 μM NE suppressed gamma power. This effect was inhibited by pretreatment with the α1A adrenoreceptor antagonist WB4101 (1 μM, Supplementary Fig. 6e–h). Therefore, Gq activation of BLA PV interneurons modulates BLA network activity, suggesting Gq neuromodulation as a potential mechanism for mediating transitions between network and behavioral states.

**Gq signaling in BLA PV interneurons promotes BLA network state switches in vivo.** Given the robust influence of Gq signaling on the pattern of BLA PV interneuron firing and the remodeling of local network activity ex vivo, we hypothesized that Gq activation in PV neurons may also reconfigure BLA population-level neural activity in vivo. To test this, we recorded LFPs in the BLA of awake PV-Cre mice with selective viral expression of hM3D-Gq (hDLX-DIO-hM3D-mCherry) in BLA PV interneurons (Fig. 6a–c). Gq activation in BLA PV interneurons with

intraperitoneal (IP) injection of 5 mg/kg (1 mg/ml) CNO significantly decreased slow and fast gamma (40–70 Hz and 70–120 Hz, respectively) oscillatory power in the BLA (Fig. 6d–f; Supplementary Fig. 8a–c). Neither saline nor CNO control injections changed BLA network activity (Supplementary Fig. 7a–c, e–g). Due to the transient nature of gamma oscillatory activity, it's possible that a large portion of the power distribution arises from background gamma activity instead of activated gamma network states[43,44]. To better isolate activated gamma states, we analyzed the upper tail of each treatment's gamma power distribution with powers greater than one standard deviation above the mean (Fig. 6g; Supplementary Fig. 8d). Comparison of power distribution tails recapitulated our initial findings, demonstrating a significant decrease in gamma power relative to baseline following hM3D activation with CNO (Fig. 6g). Similarly, neither saline nor CNO control injections changed the magnitude of gamma power distribution tails relative to baseline (Supplementary Fig. 7d, h).

We next tested whether BLA α1A adrenoreceptor activation modulates BLA network oscillations in a similar fashion using an in vivo neuropharmacological approach (Fig. 7a, b; Supplementary Fig. 9). Like PV neuron-specific Gq activation, intra-BLA infusion of NE (10 mM, 0.3 μL) reduced fast gamma power and trended

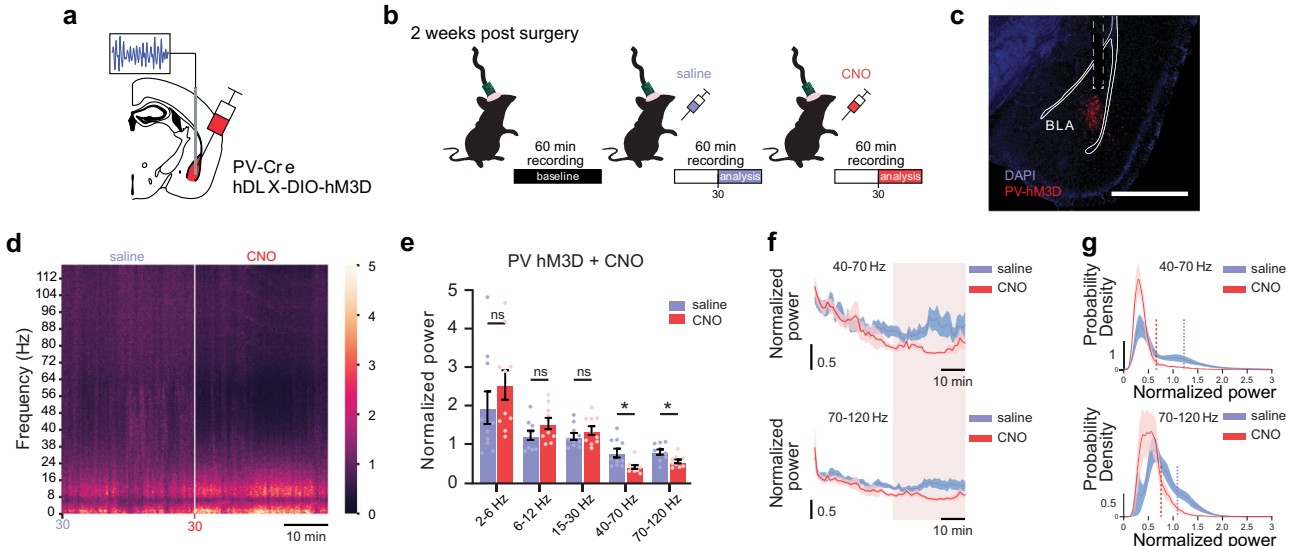

**Fig. 6 Gq signaling in BLA PV interneurons reconfigures BLA oscillatory activity in vivo. a** Schematic illustrating hDLX-DIO-hM3D-mCherry injection and LFP recording strategy in the BLA of PV-Cre mice. **b** Experimental timeline for LFP recordings before (baseline) and after IP injection of saline, followed by CNO. **c** Representative image illustrating hM3D-mCherry expression and LFP recording electrode placement in the BLA (dashed lines). Scale bar, 1 mm. **d** Average spectrogram illustrating normalized power (0–5: low-to-high power) across frequencies over the last 30 min of saline and CNO treatments. **e** Mean (±SEM) normalized power across treatments (CNO vs. saline) and frequency bands. Values from last 30 min of recording were used for analysis, illustrated by the highlighted sections in **f**. Two-way ANOVA [treatment × frequency], $F(1.081, 9.728) = 4.024$, $p = 0.0714$; Sidak's multiple comparisons test: 2–6 Hz, $p = 0.6152$; 6–12 Hz, $p = 0.4240$; 15–30 Hz, $p = 0.6838$; 40–70 Hz, $p = 0.0228$; 70–120 Hz, $p = 0.0120$. $n = 10$ mice. *$p < 0.05$, ns not significant. **f** Normalized power across time for 40–70 Hz (top) and 70–120 Hz (bottom). Values averaged in 1-min bins. Blue/red lines = average power over time, blue/red-shaded areas = SEM. Highlighted region indicates last 30 min included in analysis. **g** Average probability density plots illustrating the distribution of gamma powers across treatments for 40–70 Hz (top) and 70–120 Hz (bottom). Blue/red solid lines = average probability, blue/red-shaded areas = SEM. Vertical lines indicate average threshold value for distribution tails, where threshold = 1 standard deviation above the mean. Paired $t$-test (two-tailed), saline vs. CNO 40–70 Hz distribution tail powers ($n = 10$ mice, $p = 0.0027$); saline 2 vs. NE 70–120 Hz distribution tail powers ($n = 10$ mice, $p = 0.0005$). Source data are provided as a Source Data file.

towards suppressing slow gamma power in the BLA (Fig. 7c–e). Analysis of gamma distribution tails supported this finding and revealed a significant decrease in activated slow and fast gamma (Fig. 7f). Interestingly, intra-BLA NE infusion also potentiated theta power specifically in the low theta (2–6 Hz) band, an effect not seen with PV-specific Gq activation, which suggested that NE has the capacity to alter different oscillatory states (Fig. 7c–e). These effects were abolished by intra-BLA pretreatment with the α1A-selective adrenoreceptor antagonist WB4101 (10 μM, 0.3 μL) (Fig. 7g–j). Importantly, these data indicate that, like hM3D-Gq activation, α1A adrenoreceptor-dependent noradrenergic signaling in BLA PV interneurons reorganizes network states via an endogenous signaling mechanism in vivo.

**α1A noradrenergic Gq activation of BLA PV interneurons enhances conditioned fear expression.** Suppression of fast gamma and potentiation of theta oscillatory power among BLA networks, similar to the NE-induced changes in oscillatory states that we observed, have been reported during conditioned fear expression[7], which suggests a possible role for Gq neuromodulation of BLA PV neurons in fear memory formation. We tested the hypothesis that Gq signaling in BLA PV interneurons promotes conditioned fear expression using a virus-mediated, selective α1A adrenoreceptor re-expression strategy bilaterally in the BLA of a global α1A adrenoreceptor knockout (*Adra1a* KO) mouse model.

To confirm that selective α1A adrenoreceptor re-expression rescues NE signaling seen with endogenous α1A adrenoreceptors, we first examined NE-induced IPSC bursts using whole-cell recordings in principal neurons in brain slices from global *Adra1a*

KO mice with and without re-expression of α1A adrenoreceptors in BLA PV neurons. To selectively re-express α1A adrenoreceptors in BLA PV interneurons, a Cre-dependent AAV virus expressing α1A and mCherry (AAVdj-hDLX-DIO-α1A-mCherry) was injected bilaterally into the BLA of PV-Cre::*Adra1a* KO mice (Fig. 8a). In slices from *Adra1a* KO mice, NE and the selective α1A adrenoreceptor agonist A61603 failed to generate the phasic IPSC bursts in BLA principal cells seen in slices from wild-type mice (Fig. 8b–d; Supplementary Fig. 10a, b), confirming the specific α1A adrenoreceptor dependence of the NE-induced IPSC burst generation. Re-expression of α1A adrenoreceptors specifically in BLA PV interneurons of *Adra1a* KO mice rescued the NE-induced phasic IPSC bursts in the principal neurons; the NE-induced IPSC bursts showed the characteristic accelerating intra-burst IPSC frequency and low-frequency burst recurrence ($0.033 \pm 0.002$ Hz, $n = 31$ bursts from 7 cells, three mice) seen in principal cells in slices from wild-type mice (Fig. 8e, f).

Having demonstrated the validity of the viral re-expression model, we next investigated the role of α1A noradrenergic activation of BLA PV interneurons in the modulation of conditioned fear expression. Three weeks after α1A adrenoreceptor re-expression in the BLA with bilateral injections of AAVdj-hDlx-DIO-α1A-mCherry into the BLA of PV-Cre::*Adra1a* KO mice (Fig. 8g), the mice were subjected to a standard auditory-cued fear conditioning paradigm (Fig. 8h). Selective rescue of α1 A receptors bilaterally in BLA PV interneurons of PV-Cre::*Adra1a* KO mice resulted in enhanced fear memory retrieval on day 2 compared with PV-Cre::*Adra1a* KO mice expressing mCherry alone (Fig. 8i). Note that α1A re-expression in the PV neurons appeared to reverse a loss of fear memory retrieval in PV-Cre::*Adra1a* KO mice. The impairment in fear retrieval was presumably due to Cre and/or control virus

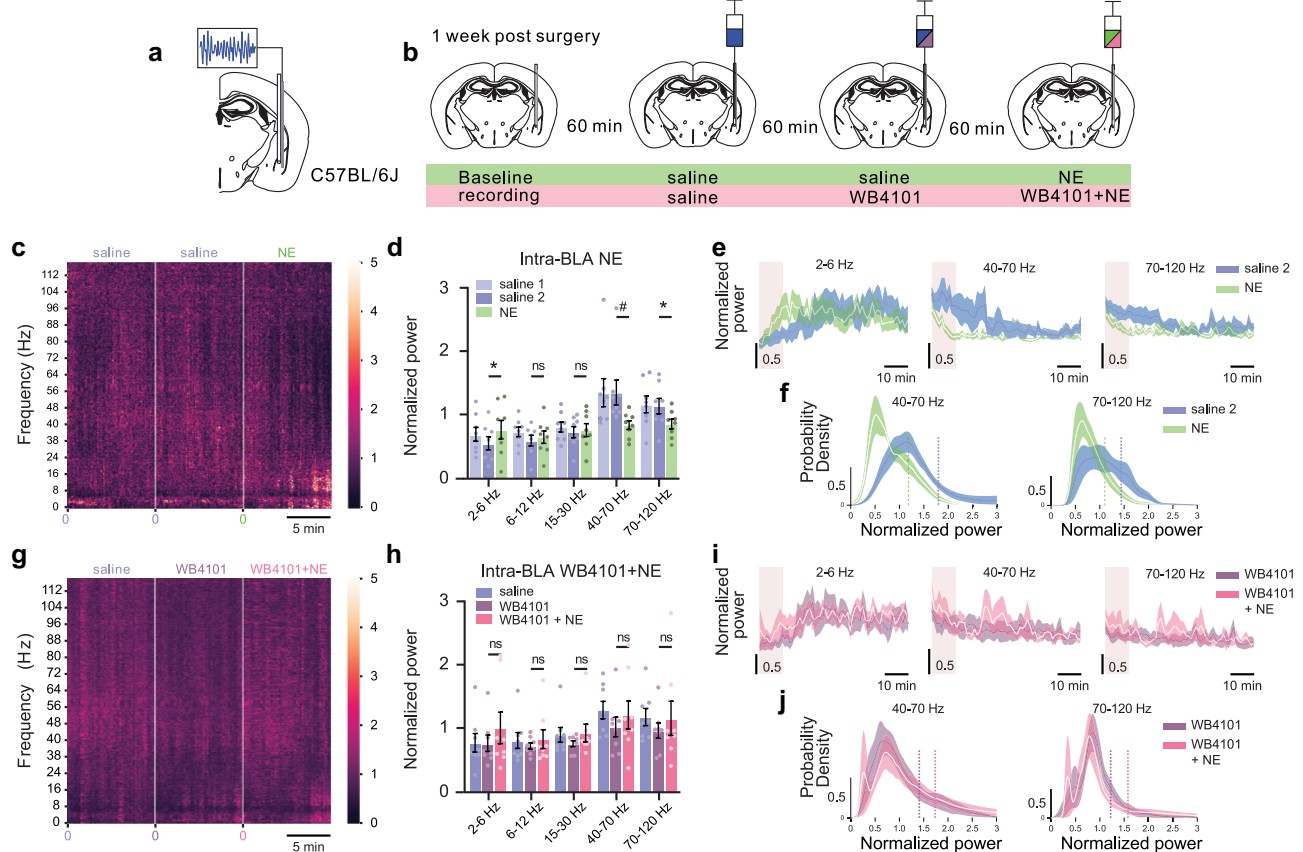

**Fig. 7 α1A adrenoreceptor signaling dynamically reorganizes BLA network states in vivo. a, b** Schematics illustrating in vivo paired intra-BLA drug microinfusion and LFP recording paradigm. **c** Average spectrogram illustrating normalized power (0–5: low-to-high power) across frequencies over the first 10 min of saline 1, saline 2, and NE treatments. **d** Mean (±SEM) normalized power across treatments (Saline 1 vs. Saline 2 vs. NE) and frequency bands. Values from first 10 min of each treatment were included for analysis. Two-way ANOVA [treatment × frequency], F(1.659, 11.61) = 5.736, p = 0.0223; Tukey's multiple comparisons test, 2–6 Hz, p = 0.0319; 6–12 Hz, p = 0.4088; 15–30 Hz, p = 0.5979; 40–70 Hz, p = 0.0784; 70–120 Hz, p = 0.0131. n = 8 mice. #p = 0.0784, *p < 0.05, ns not significant. **e** Normalized power across time for 2–6 Hz (left), 40–70 Hz (middle), and 70–120 Hz (right). Values averaged into 1-min bins. Colored lines = average power over time, colored shaded areas = SEM. **f** Average probability density plots illustrating the distribution of gamma powers across treatments for 40–70 Hz (left) and 70–120 Hz (right). Blue/green solid lines = average probability, blue/green-shaded areas = SEM. Vertical lines indicate average threshold value for distribution tails, where threshold = 1 standard deviation above the mean. Paired t-test (two-tailed), saline 2 vs. NE 40–70 Hz distribution tail powers (n = 8 mice, p = 0.0351); saline 2 vs. NE 70–120 Hz distribution tail powers (n = 8 mice, p = 0.0064). **g** Average spectrogram illustrating normalized power (0–5: low-to-high power) across frequencies over the first 10 min of saline, WB4101, and WB4101 + NE treatments. **h** Mean (±SEM) normalized power across treatments (Saline vs. WB4101 vs. WB4101 + NE) and frequency bands. Values from first 10 min of each treatment were included for analysis. Two-way ANOVA [treatment × frequency], F(2.294, 16.06) = 1.250, p = 0.3175; Tukey's multiple comparisons test, 2–6 Hz, p = 0.3855; 6–12 Hz, p = 0.7577; 15–30 Hz, p = 0.5846; 40–70 Hz, p = 0.6894; 70–120 Hz, p = 0.8026. n = 8 mice. ns = not significant. **i** Normalized power across time for 2–6 Hz (left), 40–70 Hz (middle), and 70–120 Hz (right). Values averaged into 1-min bins. Colored lines = average power over time, colored shaded areas = SEM. **j** Average probability density plots illustrating the distribution of gamma powers across treatments for 40–70 Hz (left) and 70–120 Hz (right). Blue/red solid lines = average probability, blue/red-shaded areas = SEM. Vertical lines indicate average threshold value for distribution tails, where threshold = 1 standard deviation above the mean. Paired t-test (two-tailed), WB4101 vs. WB4101 + NE 40–70 Hz distribution tail powers (n = 8 mice, p = 0.3792); WB4101 vs. WB4101 + NE 70–120 Hz distribution tail powers (n = 8 mice, p = 0.5200). Source data are provided as a Source Data file.

expression in the PV neurons and not to the loss of α1A receptors, since α1A receptor deletion in global *Adra1a* KO mice not crossed with the PV-Cre mouse did not show a decrease in fear memory retrieval compared to wild-type mice (Supplementary Fig. 10c). These data suggest that α1A adrenoreceptor signaling in BLA PV interneurons increases the expression of fear memory.

We next tested the selective activation of hM3D in BLA PV interneurons in the global *Adra1a* knockout mouse for changes in fear conditioning to determine the role of PV neuron-specific Gq modulation in fear memory formation. Three weeks after viral expression of hM3D-Gq in BLA PV interneurons with bilateral injections of AAVdj-hDlx-DIO-hM3D-mCherry or control AAVdj-hDlx-DIO-mCherry into the BLA of PV-Cre::*Adra1a* KO mice, the mice were subjected to a standard auditory-cued fear conditioning

paradigm (Fig. 8h). CNO was administered IP in Gq-DREADD and control virus-injected mice 30 min prior to the beginning of the fear acquisition trials on day one, and again 30 min prior to the beginning of the fear memory retrieval trials on day two (Fig. 8h). CNO injection enhanced fear memory retrieval in PV-Cre::*Adra1a* KO mice with selective hM3D-Gq expression in BLA PV interneurons (Fig. 8j). Therefore, our data indicate that Gq signaling in BLA PV interneurons promotes conditioned fear expression.

## Discussion
Our findings reveal a PV cell type-specific neuromodulatory mechanism for BLA network coordination of emotionally salient brain-state transitions. Activation of Gq signaling in PV

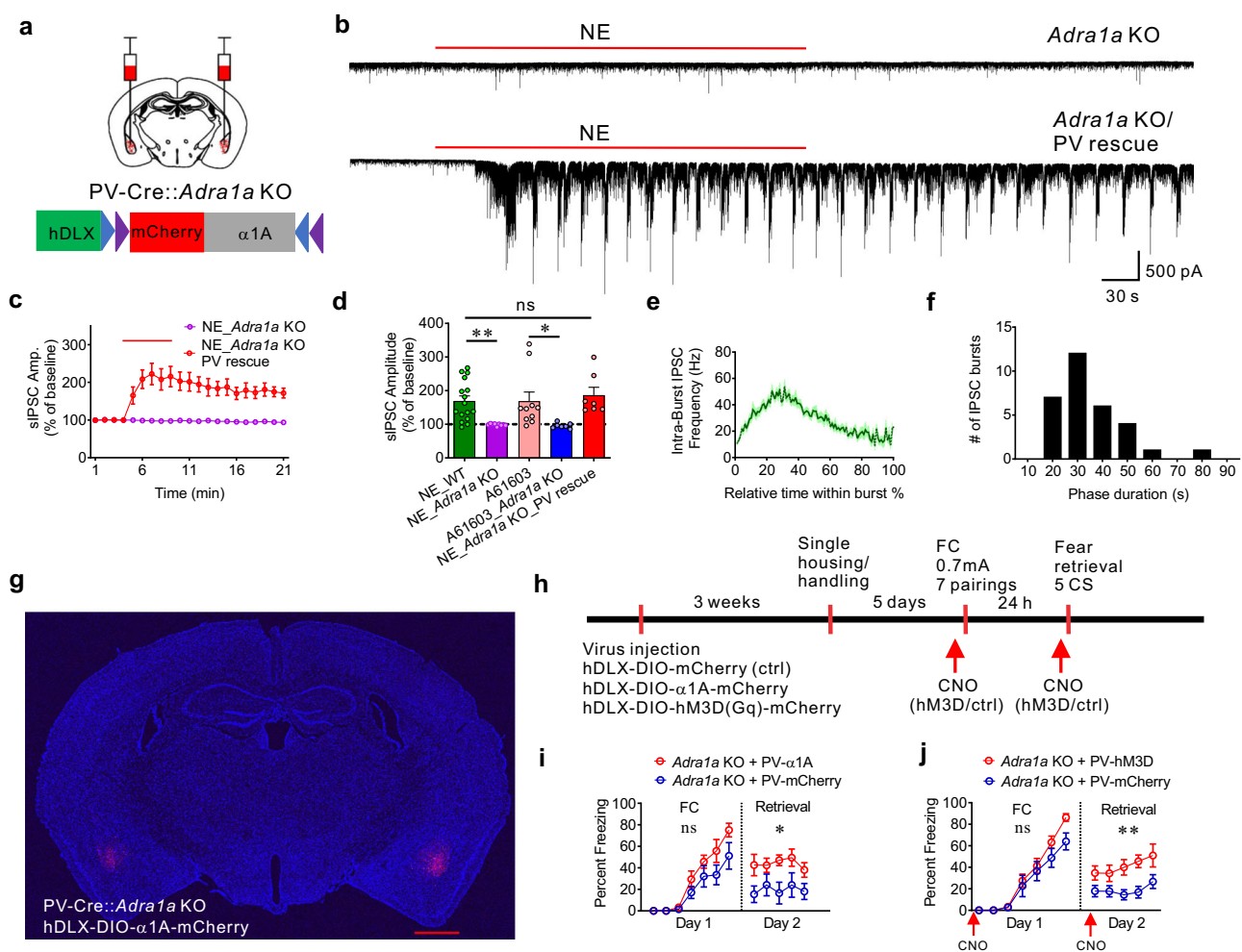

**Fig. 8 Rescue of α1A adrenoreceptor and Gq signaling in BLA PV interneurons in a global α1A receptor knockout (*Adra1a* KO) mouse facilitates fear memory formation. a** Schematic of the BLA injection sites in PV-Cre:: *Adra1a* KO mouse and AAV viral vector containing the floxed α1A, mCherry, and hDLX promoter. **b** Representative traces showing loss of NE-induced IPSC bursts in principal neurons from *Adra1a* KO mouse (top) and recovery following replacement of *Adra1a* selectively in PV interneurons (bottom). **c** NE effect on sIPSC amplitude (mean ± SEM) in principal neurons from *Adra1a* KO mice with or without re-expression of *Adra1a* in PV interneurons over time (n = 7 and 9 cells from 3 and 4 mice, respectively). **d** Mean (±SEM) changes in sIPSC amplitude in response to NE or the α1A agonist A61603 in principal neurons from wild-type (WT) and *Adra1a* KO mice. *Adra1a* re-expression in PV interneurons restored NE facilitation of sIPSC amplitude in BLA principal neurons (NE-WT, n = 16 cells from five mice; NE-*Adra1a* KO, n = 9 cells from 4 mice; A61603-WT, n = 10 cells from 4 mice; A61603-*Adra1a* KO, n = 7 cells from 3 mice; NE-*Adra1a* KO-PV rescue, n = 7 cells from 3 mice) (Unpaired *t*-test (two-tailed), A61603-WT vs. A61603-*Adra1a* KO: p = 0.039) (One-Way ANOVA, F(2, 29) = 7.43, p = 0.0025; Dunnett's multiple comparisons test, NE-WT vs. NE-*Adra1a* KO: p = 0.0046, NE-WT vs. NE-*Adra1a* KO-PV rescue: p = 0.68). **e** Mean instantaneous intra-burst IPSC frequency over the course of the burst, showing rescued IPSC bursts display an accelerating intra-burst frequency similar to WT mice (31 bursts from 7 cells, 3 mice). **f** Distribution of phase duration of NE-induced IPSC bursts following BLA *Adra1a* rescue in *Adra1a* KO mice (31 bursts from 7 cells, 3 mice). **g** α1A-mCherry expression after bilateral virus injections. Scale bar = 1 mm. **h** Timeline of virus injection, fear conditioning, CNO injections in PV-Cre::*Adra1a* KO mice. CNO (5 mg/kg, I.P.) was administrated 30 min before fear acquisition and before fear retrieval (red arrows). **i** *Adra1a* replacement in BLA PV interneurons in *Adra1a* KO mice had no effect on fear acquisition (n = 11 and n = 10 *Adra1a* KO mice with expression of α1A adrenoreceptors and mCherry in BLA PV interneurons, respectively, Mean ± SEM, Two-Way repeated measures ANOVA, F(1, 19) = 3.49, p = 0.077), but facilitated retrieval of the fear memory (Mean ± SEM, Two-Way repeated measures ANOVA, F(1, 19) = 6.92, p = 0.017). **j** Rescue of Gq signaling with excitatory DREADD in BLA PV interneurons from *Adra1a* KO mice had no effect on fear memory acquisition (n = 11 and n = 9 *Adra1a* KO mice with expression of hM3D and mCherry in BLA PV interneurons, respectively, Mean ± SEM, Two-Way repeated measures ANOVA, F(1,18) = 2.15, p = 0.16 compared to Controls), but enhanced fear memory retrieval (Mean ± SEM, Two-Way repeated measures ANOVA, F(1,18) =;11.17, p = 0.0036). **p < 0.01, *p < 0.05, ns not significant. Source data are provided as a Source Data file.

interneurons via hM3D-Gq and α1A adrenoreceptors generated a highly stereotyped phasic bursting pattern of activity that drove repetitive, synchronized bursts of IPSCs and spikes in post-synaptic BLA principal neurons. Gq-DREADD and α1A receptor activation in BLA PV neurons decreased gamma oscillatory power. Finally, BLA PV-Gq signaling enhanced retrieval of auditory cue-conditioned fear memory, consistent with the suppression of BLA gamma oscillations.

In contrast to the canonical electrophysiological property of PV interneurons to fire sustained high-frequency action potentials upon depolarization, we observed repetitive bursts of action potentials in PV interneurons in response to α1A and hM3D receptor-induced Gq activation. The PV neuron phasic bursting was dependent on a postsynaptic intrinsic Gq signaling mechanism, and not mediated by local circuits or depolarization-induced activation, since it was not suppressed by blocking

ionotropic glutamate and GABA receptors and was not induced with sustained photostimulation of BLA PV neurons. This revealed an important role for Gq neuromodulation in changing the operational mode of PV interneurons from tonic activation to alternating cycles of activation and inhibition, via an intrinsic mechanism independent of depolarization and fast chemical synaptic transmission. Further studies will be necessary to determine the signaling mechanisms downstream from Gq responsible for inducing the intrinsic oscillatory activity in the PV interneurons.

The phasic bursting activity in PV neurons was induced in this study by activation of a virally transduced Gq-coupled DREADD and by exogenous agonist application. While NE increased IPSC frequency in principal cells at 10 μM, the threshold for the generation of IPSC bursts mediated by PV neuron activation was between 10 μM and 20 μM NE. While this is high, the local concentration of endogenously released NE in or near release sites, if similar to glutamate and GABA synapses, could reach high micromolar to millimolar concentrations[45,46]. The fact that NE and the Gq-DREADD, as well as serotonin, generate a highly reproducible and stereotypic bursting pattern of activity reveals a distinct intrinsic oscillatory capacity of the PV interneurons that is capable of modulating BLA network oscillations. It remains to be determined whether endogenous neuromodulator release triggers a similar bursting pattern of activity in PV neurons of the BLA.

Parvalbumin interneurons are critically involved in the generation of gamma-frequency oscillations in the cortex and hippocampus[1,14–17]. Mechanistic studies in these structures indicate that tonic PV neuron activity provides inhibitory signaling to principal neurons that drives local gamma rhythm generation, demonstrated by phase locking of PV neuron activity to the gamma cycle[15,16] and by optogenetic activation and inhibition of PV neuron activity[14,17,18]. Like in cortical structures, PV basket cells densely target the perisomatic region and PV chandelier cells project to the axons of BLA principal neurons to tightly regulate amygdalar output[10,11,26,27,47,48]. Therefore, altering BLA PV activity patterns is likely to have a robust effect on BLA network oscillations and amygdala output. We found that Gq activation in BLA PV neurons transformed the tonic activity pattern of PV neurons into a bursting pattern of action potentials and inhibitory synaptic inputs to BLA principal cells. This activity pattern transition in turn induced a phasic pattern of activation in local principal cells. Consistent with tonic PV neuron activity promoting gamma oscillations, transforming PV neuron tonic activity to phasic activity by either chemogenetic manipulation or α1A adrenoreceptor activation decreased gamma oscillatory power locally in the BLA both ex vivo and in vivo.

Previous work also demonstrated the capacity of BLA PV interneurons to modulate local theta-frequency network activity[10,11], and that BLA theta rhythms play a central role in facilitating conditioned fear behavior[7,9–11]. Interestingly, though interneuronal α1A receptor activation potentiated BLA theta power, BLA PV-specific Gq activation did not. This suggests that another interneuron subtype in the BLA may contribute to the theta-frequency oscillations. Indeed, BLA CCK interneurons also express α1A adrenoreceptors and may work alone or with local PV interneurons to drive the observed potentiation of 2–6 Hz power with norepinephrine infusions. These findings raise the possibility that different interneuron populations in the BLA coordinate unique oscillatory states. Our finding that NE activates inhibitory synaptic inputs to the BLA principal cells from other GABA interneuron subtypes supports this possibility. Indeed, we found that inhibiting calcium-dependent release from PV and CCK neuron axons blocked ~50% of the NE-induced increase in sIPSC frequency (Supplementary Fig. 3), suggesting that the

remaining ~50% of the NE response was generated by one or more other interneuron subtypes, such as the somatostatin-expressing inhibitory interneurons. Interestingly, the response was blocked completely by an α1A receptor antagonist (Supplementary Fig. 3), was insensitive to β adrenoreceptor blockade (Supplementary Fig. 3), and was entirely lost in the α1A receptor knockout mouse (Supplementary Fig. 10), which indicated that the NE stimulation of GABA neurons in the BLA is mediated exclusively by α1A receptors. The relative effects of synaptically released NE on different inhibitory interneuronal circuits are not known and requires further study.

Parvalbumin interneurons in the BLA target the perisomatic region of hundreds of postsynaptic principal neurons to control BLA output. Paired recordings from neighboring principal neurons revealed that a large percentage of the PV-mediated bursts of IPSCs are synchronized. In addition, PV-mediated IPSC bursts transformed the firing pattern of local principal neurons from a tonic to slow phasic activity. However, other than the disruption of gamma wave generation, it is not known whether the slow synchronous phasic PV neuron inhibitory inputs to BLA principal neurons play a physiological role in pattern generation in BLA circuits. Further, it remains to be determined in what contexts NE induces changes in PV interneuron activity and transitions between network states in the BLA, although converging evidence suggests a potential role in the behavioral expression of fear.

Given the influence of BLA noradrenergic signaling on promoting fear states[49], we examined what significance the PV neuron α1A adrenoreceptor-mediated reorganization of BLA network activity has in fear conditioning. Genetic re-expression of α1A adrenoreceptors and hM3D-Gq activation in PV interneurons bilaterally in the BLA of *Adra1a* global knockout mice enhanced conditioned fear expression. Therefore, the Gq-mediated transition in BLA PV interneuron patterned output appears important for promoting BLA fear state transitions. Conditioned fear expression is associated with BLA potentiation of theta and suppression of fast gamma oscillatory power[7]. Our data demonstrating that Gq signaling via α1A adrenoreceptor and hM3D receptor activation in BLA PV interneurons desynchronizes gamma oscillations and may work cooperatively with other local interneurons (e.g., CCK interneurons) to potentiate theta network activity, reveal a potential critical role for Gq neuromodulation of BLA PV cells in the expression of fear memory.

Thousands of G protein-coupled receptors converge onto four main classes of known Gα proteins[50]. In addition to α1A adrenoreceptor-mediated Gq activation of PV interneurons, we also found that Gq-coupled 5-HT2A serotonergic receptors generated a similar pattern of repetitive bursts of IPSCs in BLA principal neurons. Others have reported that acetylcholine also drives repetitive IPSC bursts in pyramidal neurons of the frontal cortex, possibly through the activation of Gq-coupled M1 muscarinic receptors[51], although whether this response is also mediated by PV interneuron activation is not known. Notably, PV interneurons in the hippocampus and BLA stimulated by CCK activation of Gi/o-coupled CCK_B receptors[52–55] generate tonic IPSCs in principal neurons. Thus, different neurotransmitters acting on their cognate Gq-coupled receptors have a similar effect on neural activity through activation of the same signaling pathway, whereas the neuromodulatory excitation of PV interneurons via different GPCR signaling mechanisms induces distinct output patterns. If the patterned bursts of IPSCs is a generalized PV neuron output mode following activation of Gq-coupled receptors by different neuromodulators, then the specificity of the neuromodulatory regulation must lie in other spatiotemporal properties of the neuromodulatory inputs, such as the timing of the inputs and the somato-dendritic distribution of

the synapses or synaptic receptors. Then again, the convergence of disparate neuromodulatory inputs onto the Gq signaling pathway to generate a similar patterned PV neuron output may represent a redundancy of neuromodulatory systems to achieve a common salient network outcome, the desynchronization of BLA circuits in the gamma-frequency band, under different environmental conditions. Just as different representations of intrinsic membrane ionic currents in individual neurons and synaptic strengths in neural circuits can be tuned homeostatically to achieve target network outputs[56], so may neuromodulatory inputs be activated variably, alone or in combination, to arrive at a particular desired brain state.

In conclusion, our findings provide a cell type-specific neuromodulatory mechanism for BLA network-driven transitions in fear-associated brain states. These data suggest that during conditioned fear recall, BLA PV neurons transition from a tonic, high-frequency pattern to a phasic bursting pattern of activity in response to Gq activation, which desynchronizes gamma oscillatory activity in the BLA and facilitates the expression of conditioned fear memory. This study demonstrates, therefore, the capacity of neuromodulators to control network and behavioral state transitions.

## Methods

**Animals**. Mice were maintained in AALAC-approved, temperature-controlled animal facilities on a 12-h light/dark cycle with food and water provided *ad libitum*. C57BL/6 J (Cat. 000664), PV-Cre (Cat. 017320), Ai14 (Cat. 007914), and *Adra1a* KO mice (Cat. 005039) were purchased from Jackson Laboratories and bred in-house to establish colonies. Heterozygous GAD67-eGFP mice were purchased from Riken BioResource center[37] and back-crossed for >5 generations with wild-type C57BL/6 mice. All procedures were approved by the Tulane University or the Tufts University Institutional Animal Care and Use Committees and were conducted in accordance with Public Health Service guidelines for the use of animals in research.

### Stereotaxic surgery

*Intracerebral virus injections*. Four-to-six-week-old male mice were anesthetized by intraperitoneal injection of ketamine/xylazine (100 mg/kg) and placed in a stereotaxic frame (Narishige, SR-6N). The scalp was cut along the midline and the skull was exposed. Two burr holes were drilled above the BLA with a Foredom drill (HP4-917). Mice were then injected bilaterally with 350 nL of the virus into the BLA (AP: −0.8, ML: 3.05, DV: 4.4) through a 33-gauge Hamilton syringe (10 µl) connected to a micropump (World Precision Instrument, UMP-2) and controller (Micro4) at a flow rate of 100 nL per minute. After waiting for 5 minutes following virus injection to minimize virus spread up the needle track, the injection needle was then slowly retracted from the brain. After surgery, the scalp was sealed with Vetbound, a triple antibiotic ointment was applied, and an analgesic (Buprenorphine, 0.05 mg/kg) was injected IP.

### Extracellular recording and drug application

Eight to ten-week-old male mice were anesthetized with IP ketamine/Xylazine (100 mg/kg; 10 mg/kg) and placed in a mouse stereotaxic frame (World Precision Instruments, 502600) over a warm heating pad. Lacri-lube was placed over the subjects' eyes, and slow-release buprenorphine (Buprenorphine SR-LAB, 0.5 mg/kg) was administered subcutaneously for post-operative analgesia. The scalp was shaved, cleaned with ethanol and betadine (3x), then cut along the midline to expose the skull. The skull was leveled, then manually drilled above the BLA (relative to bregma: AP −1.35, ML ± 3.3) using a sterile 27 G syringe needle. In dual-electrode and intra-BLA cannula implantation surgeries, a second hole was manually drilled at AP −3.05, ML ± 3.3. For viral injections, a pulled glass pipette was used to inject 350 nL of the virus into the BLA (relative to bregma: AP −1.35, ML ± 3.3, DV −5.1) at a flow rate of 100 nL per minute using positive pressure from a 10 mL syringe. After waiting 5 min following injection to minimize viral spread up the needle track, the injection needle was then slowly retracted from the brain. For LFP electrode implants, prefabricated headmounts (Pinnacle Technology Inc, #8201) were mounted to the skull using stainless-steel screws that acted as ground and reference, while trying to avoid damaging the cortex. The BLA depth electrode (PFA-coated stainless steel, A-M Systems) was implanted using the same coordinates as for virus injection (relative to bregma: AP −1.35, ML ± 3.3, DV −5.1). For paired intra-BLA cannula and electrode implantation, the recording equipment was implanted and fixed to the skull as described. A fabricated guide cannula was inserted over the BLA, through the posterior drilled hole (relative to bregma: AP −3.05, ML ± 3.3, relative to skull: −3.08) at a 69° angle (Supplementary Fig. 9). The cannula was fixed in place with dental cement (A-M Systems, #525000 and #52600)

before release from the stereotaxic arm. After surgery, the scalp was either (1) sealed with Vetbond and triple antibiotic ointment was applied or (2) covered along with the headmount/guide cannula with dental cement and allowed to cure before removing the mouse from the stereotaxic frame. The mice were then taken out of the stereotaxic frame and placed in a heated recovery chamber until conscious.

**AAV virus development**. For cloning of Cre-dependent hDlx AAV virus, we amplified the hM3D-mCherry and mCherry coding sequences from the plasmid pAAV-hSyn-DIO-hM3D(Gq)-mCherry (a gift from Bryan Roth, Addgene # 44361)[57] and pAAV-hSyn-DIO-mCherry (also a gift from Bryan Roth, Addgene # 50459) and cloned them into a pAAV-hDlx-Flex-GFP vector backbone (a gift from Gordon Fishell, Addgene # 83895)[58] at AccI and NheI cloning sites. The coding sequence of *adra1A* was synthesized from Bio Basic Inc. and cloned into a pAAV-hDlx-Flex backbone. The virus was further packaged by Vigene Biosciences Company in AAVdj serotype. pAAV-EF1a-DIO-ChR2-mCherry plasmid (a gift from Karl Deisseroth, addgene # 20297) was purchased from addgene and packaged into AAVs from Vigene Biosciences Company in serotype9. All hDlx AAV viruses were diluted to the range of $10^{11}$ to $10^{12}$ viral genome per ml with virus dilution buffer containing 350 mM NaCl and 5% D-Sorbitol in PBS.

**Fear conditioning and retrieval**. Three weeks after virus injection, male mice were single housed and handled for >5 days before undergoing a fear conditioning paradigm with the Video Fear Conditioning System in a sound attenuated chamber (MED Associates, Inc.). Each chamber is equipped with a metal stainless-steel grid connected to a shock generator (ENV414S Aversive Stimulator). The fear conditioning paradigm consisted of seven exposures to a continuous tone (7 kHz, 80 db, 30 s duration) as the conditioned stimulus (CS), each of which was co-terminated with an unconditioned aversive stimulus (US) consisting of an electric foot shock (0.7 mA, 2 s duration). The CS-US stimuli were presented at a randomized intertrial interval (ITI, 30–180 s, average = 110 s) in one context, context A. Twenty-four hours later, on day 2, mice were tested for fear retention in a different context, context B, with a planar floor and a black plastic hinged A-frame insert. During fear memory retrieval, five presentations of CS alone were delivered with an inter-stimulus interval of 60 s. Behavior was recorded with an infrared camera and analyzed with Video Freeze software (Med Associates, Inc.). Mice were considered to be exhibiting freezing behavior if no movement other than respiration was detected for ≥2 s. Chambers were cleaned with either 70% ethanol or 3% acetic acid before each session of fear conditioning and fear memory retrieval.

### Histology

*Perfusion and cryosectioning*. Two weeks after AAV virus injection, *Adra1a* KO, PV-Cre, and PV-Cre::*Adra1a* KO were deeply anesthetized with ketamine/xylazine (300 mg/kg) and perfused transcardially with 10 ml ice-cold PBS (pH 7.4) followed by 20 ml 4% paraformaldehyde (PFA) in PBS. Brains were dissected out, postfixed for 3 h in 4% PFA in PBS, and cryopreserved with 30% sucrose in PBS for 24 h at 4 °C until the brain sunk to the bottom of the container. Coronal sections (45 µm) were cut on a cryostat (Leica) and harvested in 24-well plates filled with PBS.

*Confocal imaging*. Sections from virus-injected PV-Cre and PV-Cre::*Adra1a* KO mice containing the BLA were selected, rinsed with PBS (3 × 5 mins), and mounted on gel-coated slides. Confocal images were acquired with a Nikon A1 confocal microscope to capture the DAPI (excitation 405 nm, emission 450 nm), GFP (excitation 488 nm, emission 525 nm), and mCherry (excitation 561 nm, emission 595 nm) signals. For the analysis of colocalization, z-stack pictures were imaged under a 40x oil-immersion objective with a step increment of 1.5 µm. The number of BLA cells containing colocalized markers was first quantified from merged maximal intensity images from different channels, and then confirmed in z-stack images with ImageJ (NIH).

*X-gal staining*. Sections from *Adra1a* KO mice were rinsed with PBS (3 × 5 min) and incubated in a β-gal staining solution (Roche, Ref # 11828673001) overnight. After β-gal staining, sections were then rinsed in PBS (3 × 5 min), mounted on gel-coated glass slides, coverslipped with Permount mounting medium (Fisher Scientific), and allowed to air dry. Bright-field imaging was performed in a Zeiss Axio Scanner and processed and analyzed with ImageJ (NIH). For fluorescence confocal imaging, brain sections were rinsed with PBS (3 × 5 min), mounted on gel-coated coverslips, and then imaged first on the confocal microscope (Nikon A1) before incubating them in the β-gal staining solution. After staining with X-gal, the slices were rinsed with PBS (3 × 5 mins) and the same regions previously imaged using fluorescence confocal imaging were then re-imaged for the X-gal signal with excitation and emission wavelengths of 638 nm and 700 nm[38], respectively. The images were then processed and quantified with ImageJ software to determine the ratio of β-gal-positive cells to fluorescent cells with the same procedure as described above.

### Brain slice electrophysiology

*Slice preparation*. Coronal brain slices containing the BLA were collected from 6–10-week-old mice for ex vivo patch clamp and extracellular field recordings.

Mice were unanesthetized (for patch clamp recordings) or anesthetized (for field potential recordings) with isoflurane and decapitated in a restraining plastic cone (DecapiCone, Braintree Scientific) using a rodent guillotine. Brains were extracted and immersed in ice-cold (0–4 °C) sucrose cutting solution containing (in mM) 150 or 252 sucrose (for field or patch recordings), 33 or 0 NaCl (for field or patch recordings), 25–26 NaHCO$_3$, 2.5 KCl, 1.25 NaH$_2$PO$_4$, 1 CaCl$_2$, 7 or 5 MgCl$_2$ and 15 or 10 glucose (300–310 mOsm), which was oxygenated with 95% O$_2$ and 5% CO$_2$. The brains were trimmed and coronal brain slices (300–350 μm) were sectioned using a vibratome (VT1000s or 1200, Leica). After sectioning, slices for patch clamp recordings were transferred to a holding chamber containing oxygenated patch clamp recording medium (aCSF$_{patch}$) containing (in mM): 126 NaCl, 2.5 KCl, 1.25 NaH$_2$PO$_4$, 1.3 MgCl$_2$, 2.5 CaCl$_2$, 26 NaHCO$_3$, and 10 glucose, where they were maintained at 34 °C for 30 min before decreasing the chamber temperature to ~20 °C. Slices for field recordings were transferred after sectioning to a petri dish containing the sucrose cutting solution, where they were trimmed to remove inputs from the hippocampus, and then transferred to an interface holding chamber filled with recording medium (aCSF$_{field}$) continuously oxygenated with 95% O$_2$ and 5% CO$_2$ and containing (in mM): 126 NaCl, 10 glucose, 2 MgCl$_2$, 2 CaCl$_2$, 2.5 KCl, 1.25 NaH$_2$PO$_4$, 26 NaHCO$_3$, 1.5 Na-pyruvate, 1 L-glutamine (300–310 mOsm) set to 34 °C. All slices were incubated for at least 1 h before being transferring to a submerged recording chamber containing aCSF$_{patch}$ for patch clamp recordings or to an interface recording chamber containing aCSF$_{field}$ for field recordings.

*Patch clamp recording.* Slices were bisected down the midline and hemi-slices were transferred one at a time from the holding chamber to a submerged recording chamber mounted on the fixed stage of an Olympus BX51WI fluorescence microscope equipped with differential interference contrast (DIC) illumination. The slices in the recording chamber were continuously perfused at a rate of 2 ml/min with recording aCSF$_{patch}$ maintained at 32-34 °C and continuously aerated with 95% O$_2$/5% CO$_2$. Whole-cell patch clamp recordings were performed in putative principal neurons in the BLA. Glass pipettes with a resistance of 1.6-2.5 MΩ were pulled from borosilicate glass (ID 1.2 mm, OD 1.65 mm) on a horizontal puller (Sutter P-97) and filled with an intracellular patch solution containing (in mM): 110 CsCl, 30 potassium gluconate, 1.1 EGTA, 10 HEPES, 0.1 CaCl$_2$, 4 Mg-ATP, 0.3 Na-GTP, 4 QX-314; pH was adjusted to 7.25 with CsOH and the solution had a final osmolarity of ~290 mOsm. DNQX, APV, TTX, Prazosin, Propranolol, WB4101, A61603, CNO, and NE were delivered at the concentrations indicated via the perfusion bath. Slices were pre-incubated in aCSF$_{patch}$ containing ω-agatoxin (0.5 μM, 30 min), ω-conotoxin (0.5 μM, 30 min), or YM-254890 (10 μM, 20 min) to block P/Q-type calcium channels, N-type calcium channels, and Gα$_{q/11}$ activity, respectively[25,32]. The same solution as that used for the aCSF$_{patch}$ was used in the patch pipettes (1.6-2.5 MΩ) for loose-seal patch clamp recording of action potentials, which were performed in the $I = 0$ mode on the patch clamp amplifier. For current-clamp recordings, an intracellular patch solution was used that contained (in mM): 130 potassium gluconate, 10 HEPES, 10 phosphocreatine Na$_2$, 4 Mg-ATP, 0.4 Na-GTP, 5 KCl, 0.6 EGTA; pH was adjusted to 7.25 with KOH and the solution had a final osmolarity of ~290 mOsm. Series resistance was normally below 10 MΩ immediately after breaking through the membrane and was continuously monitored. Cells were discarded when the series resistance exceeded 20 MΩ. Data were acquired using a Multiclamp 700B amplifier, a Digidata 1440 A analog/digital interface, and pClamp 10 software (Molecular Devices). Recordings were filtered at 2 KHz for IPSC recordings and at 10 KHz for action potential recordings and sampled at 50 KHZ. Data were analyzed with MiniAnalysis software (SynaptoSoft, NJ) and Clampfit 10 (Molecular Devices). The interspike interval (ISI) histograms for each PV cell were plotted with a bin size of 1 ms and the predominant firing rates were derived by finding the peak ISI from the distribution. The ISI coefficient of variation was calculated by dividing the standard deviation of the ISI over the mean ISI[40]. Statistical comparisons were conducted with a paired or unpaired Student's t-test or with a one- or two-way ANOVA followed by a *post hoc* Tukey's test as appropriate ($p < 0.05$ with a two-tailed analysis was considered significant).

*Field potential recording.* BLA sections were placed in an interface recording chamber containing aCSF$_{field}$ and a borosilicate glass extracellular field recording electrode was placed in the BLA. LFP data were acquired with LabChart (ADInstruments) at 10 KHz and low-pass filtered at 3 KHz during acquisition. Gamma oscillations were induced by continuous perfusion of oxygenated modified aCSF$_{field}$ with elevated potassium (7.5 mM) and Kainic acid (800 nM) at a rate of 1.8 mL/min. Baseline and treatment field potentials were recorded for 15 min each. For DREADD experiments, only values from the last 5 min of each condition were used for analysis. For NE and WB4101 experiments, values from the last 5 min of either baseline or WB4101 were compared against the 3$^{rd}$–8$^{th}$ min of NE (roughly capturing the peak of the NE-induced gamma suppression). All values were normalized to their 15-min averaged baseline recording. Recording orders are as follows. NE experiments: (1) aCSF$_{field}$ only, (2) 100 μM NE in aCSF$_{field}$. WB4101 experiments: (1) aCSF$_{field}$ only, (2) 1 μM WB4101 in aCSF$_{field}$, (3) 1 μM WB4101 + 100 μM NE in aCSF$_{field}$.

*In vivo electrophysiology.* LFP recordings were performed in awake, freely behaving C57BL/6 J and PV-Cre mice, acquired using prefabricated headmounts (Pinnacle

Technology, #8201). BLA LFP recordings were acquired through an insulated LFP depth electrode implanted in the ipsilateral BLA. Mice were tethered to the apparatus and LFPs were recorded at 4 KHz and amplified 100x. All mice were left to habituate to the recording chamber for at least 30 minutes while tethered before recording. In PV-Cre mice expressing BLA PV hM3D, baseline and treatment conditions (I.P. injection of saline and CNO (5 mg/kg (1 mg/ml), dissolved in saline)) were recorded for 60 min each, and only the last 30 min were used for analysis given the IP route of exposure. In cannulated C57 mice, baseline and treatment conditions (intra-BLA infusion of saline, NE, WB4101, and WB4101 + NE) were recorded for 60 min each, only the first 30 min are shown, and only the first 10 min were used for analysis given the infusion route of exposure. All values, regardless of the experiment, were normalized to their 60-min averaged baseline recording.

*Extracellular field potential data analysis (in vivo and ex vivo).* LFP data were band-pass filtered (1–300 Hz, Chebyshev Type II filter), and spectral analysis was performed in MATLAB using publicly accessible custom-made scripts developed in the Maguire lab (https://github.com/pantelisantonoudiou/MatWAND) utilizing the fast Fourier transform[59]. Briefly, recordings were separated into 5 s bins with 50% overlapping segments to capture at least 10 cycles for each frequency assessed. The power spectral density for positive frequencies was obtained by applying a Hann window to eliminate spectral leakage. The mains noise (58–60 Hz band) was removed from each bin and replaced using the PCHIP method. Values 3x larger or smaller than the median were considered outliers and replaced with the nearest bin. Processed spectral data were then imported to Python for resampling into one-minute bins and normalization to baseline. Finally, normalized, resampled data were imported to GraphPad Prism for statistical analysis. Statistical comparisons were conducted using either two-tailed Student's paired t-tests or two-way ANOVAs. ANOVA multiple comparisons were corrected for using Tukey's and Sidak's multiple comparison tests as appropriate ($p < 0.05$ was considered significant). Importantly, an additional analysis using a 0.25-s window, which captured 10 cycles of the slowest gamma frequency (40 Hz), produced results similar to the 5-s window analysis (which captured 10 cycles of the slowest theta frequency (2 Hz)) (Supplementary Fig. 8).

*Cannula fabrication and intra-BLA infusions.* Intra-BLA drug infusion cannulas were fabricated in-house using 23 G and 30 G syringe needles. Twenty-seven gage syringe needles were cut at each end to produce a 10 mm plastic base and 5 mm barrel. Thirty gage syringe needle barrels were cut to 16 mm from the plastic base (including the 1 mm bit of adhesive at the base of the barrel) to create the internal cannula and inserted through the guide cannula to protrude an extra 1 mm. To produce clean syringe barrel openings, barrels were initially cut an extra 1–2 mm longer and shaved back to the desired length using a Dremel rotary tool (Dremel 4000) with a 120-grit circular sanding attachment.

Intra-BLA infusions were performed using a 25 μL Hamilton syringe affixed to an automated microinfusion pump (Harvard Apparatus, The Pump 11 Elite Nanomite), and connected to the internal cannula needle via plastic tubing (Tygon flexible plastic tubing; ID = 0.020 IN, OD = 0.060 IN). Intra-BLA infusions of 300 nL norepinephrine (10 mM), WB4101 (10 μM), or saline were administered at a rate of 0.2 μL/min. After infusion, the needle was left to sit for an extra minute past the infusion to allow for sufficient diffusion and minimize backflow upon removal from the brain.

**Statistics and reproducibility**. Imaging was repeated independently with similar results in Figs. 1b; 4a, b, d; 6c; 8g and Supplementary figure 9c.

**Reporting summary**. Further information on research design is available in the Nature Research Reporting Summary linked to this article.

## Data availability

Source data are provided with this paper. Further data supporting the findings are available upon request to the corresponding author.

## Code availability

Code used for spectral analysis of LFP data in this study is available at: https://github.com/pantelisantonoudiou/MatWAND.

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

## Acknowledgements

We thank Dr. Laura Harrison for her invaluable input in the study. We also thank India Pursell and Hunter Bernstein for their technical assistance. This work was supported by

NIH grants R01MH104373 and R01MH119283 (J.G.T); R01AA026256, R01NS105628, and R01NS102937 (J.M.), and R01MH122561 (J.P.F.).

## Author contributions

Conceptualization, J.G.T., X.F., E.T., and J.M.; investigation, X.F., E.T., G.L.W., P.A., and C.D.B.; writing—original draft, X.F. and E.T.; writing—review and editing, J.G.T., J.M., and J.P.F.; funding acquisition, J.G.T., J.M., and J.P.F.

## Competing interests

J.M. has a sponsored research agreement and serves as a member of the Scientific Advisory Board for SAGE Therapeutics, Inc. The remaining authors declare no competing interests.
