## [Peer review file · Nature Communications]

REVIEWER COMMENTS

Reviewer #1 (Remarks to the Author):

The manuscript by Fu et al. identifies that activation of Gq via DREADD or alpha1A adrenoreceptors induces an intrinsic burst firing mode in BLA PV neurons, leading to synchronized inhibitory drive in BLA principal neurons, changes in the local LFP and enhanced fear retrieval in a loss-of-function mouse line. The findings are highly interesting. However, the strong conclusions in relation to the in vivo function of this cellular burst phenotype are currently somewhat weakly supported by the data. Furthermore, some controls are missing.

1. Fig. 1C: The GAD-GFP label is unclear. Is this a mouse line, viral expression, antibody stain?
2. Fig. 1E, Fig. 3E,J: Given that the bursts have different lengths and IPSC numbers, wouldn't it be more intuitive to plot the Intra-burst IPSC frequency vs. the relative burst length? This will make the data more comparable across cells and events.
3. Fig. 1K: Vertical units / scale bar missing.
4. Fig. 1I-K: Results are based on one example cell. Was this reproducible? If yes, summary data is missing.
5. Fig. 1L: Axis units are missing.
6. Fig. 2B: Classification of synchronized events is not clear. The authors should also provide temporally expanded representative traces to illustrate the event synchrony.
7. Fig. 3 I: Units / scale bars are missing.
8. Fig. 5, 6, 7: These experiments miss controls (e.g., mCherry expressing animals similar to the controls in Fig. 8). It can't be ruled out that the observed effects are mediated by CNO alone (or drift) irrespective of DREADD expression.
9. Fig. 5: Quantification across cells should be provided.
10. Fig 6, 7: The LFP analysis window size is 5s. This seems long and the window size for the time frequency analysis should be adjusted to a specific frequency (e.g. 5-10 cycle of an oscillation to estimate theta, gamma separately). The quantification of "normalized power area" is not clear. A wavelet spectrogram would help to demonstrate the change of power in different frequency bands. Comparing the mean power of gamma oscillations in a 30 min window is problematic due to the transient nature of gamma events (alternative approaches: changes in the tail of the power distribution, bootstrapping).
11. Fig 6, 7: Theta events are state dependent. Did the analysis of theta power account for the behavioral state (eg. similar running speeds, active vs. inactive periods). Drug-related changes in behavioural state might confound the LFP analysis and should be ruled out.
12. Fig. 8 / page 23. l. 348-354: The behavioural data does not seem supportive of the proposed conclusion, i.e., that neuromodulation induced burst firing induces 'transition between ... behavioural states'. The authors speculate that the loss of fear retrieval is due to the PV-cre line, but not the general loss of alpha1A receptors. However, the PV-cre line has been successfully used in fear learning paradigms before and rescuing fear retrieval in this learning-deficient intercross line does not provide a convincing mechanistic link of transitions of behavioural states and burst firing in BLA PV neurons. Demonstrating that CNO induced burst firing in PV-neurons is correlated with enhanced fear retrieval

would be an alternative, more convincing experiment. In addition, the data that the alpha1A KO mouse has no learning deficit should be shown.

13. Fig. 8H: The time line suggests that only DREADD animals were injected with CNO. However, the mCherry animals should have been injected as well, otherwise they are not controlling for potential CNO effects. This should be clarified or the appropriate controls should be added.

14. It is not clear how general Gq activation leads to intrinsic burst firing in PV neurons. Potential mechanisms should be discussed.

Reviewer #2 (Remarks to the Author):

In this study the authors examine the impact of Gq signalling in in the basolateral amygdala. Using a variety of techniques and transgenic lines they show that activation of Gq receptors in Parvalbumin-expressing interneurons induces a stereotypical pattern of bursting electrophysiological activity. This results in patterns of bursting inhibitory input to glutamatergic principal neurons, converting activity of these neurons to a bursting oscillatory pattern. This cellular activity is seen in the local field potential as a change in in gamma band oscillations. Also using transgenic mouse lines and pharmacology they show that in the BLA this modulation can be mediated by activation of alpha1A adrenoceptor receptors. Finally, they show that this Gq activity modulates the expression of conditioned fear. The experiments in this paper are complex and provide a large body of data that convincingly backs the above conclusions. I congratulate the authors on their very well thought out and complex experiments. The data shown is clear and the conclusions they reach well supported by the data. That said, I have several issues that should be addressed.

Firstly, in almost all the experiments shown Gq receptors are activated by bath application of CNO or norepinephrine coupled with expression of different constructs in PV-interneurons (using a PV-cre line crossed with other transgenic lines). The mechanism of how the discharge pattern of these interneurons changes is not addressed but this is not the point of the paper. This method activates either adrenergic receptors on all PV-interneurons or activates drives Gq widely. Perhaps as expected, bath application of serotonin, also coupled to Gq receptors can reproduce the effects of norepinephrine. However, in vivo, the receptors are driven by transmitter release from ascending noradrenergic systems. It is not at all clear what the impact of this input would have on network activity - where there is some diffuse transmission, much of the impact of these transmitters is synaptic over short periods of release. It is not clear if any of the network activity would be seen when these systems are active. Note that the impact of norepinephrine input to the BLA on fear expression has been shown by Uematsu et al (Nature Neuroscience 20, 1602-1611

2017) and the data here (Figure 8) reproduces their findings. Surprisingly this paper is not cited. If this input to the BLA also changes PV cell activity and the activity of principal neurons is not known. While the behavioural experiments do go some way to addressing this point, it would have been very useful to test the impact of synaptically released transmitter. One option is to use a TH-cre line and optogenetics to directly release NE into the BLA. While this is not a major issue with the data shown in this study - I feel that it should be addressed in the discussion.

Secondly, all manipulations have been done using PV-cre mice. There is no doubt that PV cells are the main basket interneurons in the BLA. However, there is another class of axo-axonic interneurons in the BLA that also innervate many hundreds of principal cells. The procedures used here do not differentiate these two types. However, the manuscript is written entirely from the perspective of PV-basket cells - this separation needs to be made clear and some evidence provided for the role of basket interneurons. They do address the issue of CCK interneurons and allude to SOM-interneurons but not the axo-axonic cells. This is an important distinction.

Minor:

For the LFP data - it would be good show some straightforward power-spectra for the non-afficianado.

I was very surprised by the very high concentration of compounds used - norepinephrine up to 100 μ M while the EC50 for these compounds acting at G-protein coupled receptors is in the low micromolar range. While the pharmacological assessment is fine - there is always a change of off-target actions with these high concentrations.

Reviewer #3 (Remarks to the Author):

In this study Fu et al. present very interesting findings on the electrophysiological and behavioral impact of PV interneurons activation through GPCR Gq protein activation (exogenous DREADDs and endogenous adrenergic receptors) in the basolateral amygdala (BLA). The results provide a potential mechanism for the role of the adrenergic-2A receptor in fear encoding in the BLA, and might have a strong impact on the field of learning and memory. However, I have concerns regarding the description and interpretation of the DREADDs experiments, and other points.

First, the authors perform manipulation of exogenous Gq receptors (DREADDs), which is very different than increasing the activity of endogenous Gq receptors. The whole manuscript needs to make this point very clear, and the results should be interpreted as such. Indeed, the DREADD activation does not demonstrate that endogenous Gq activation in PV interneurons induces bursting of these neurons, as well as IPSC bursts in pyramidal neurons. The DREADDs experiments only suggests that endogenous Gq activation induces bursting in BLA PV interneurons and pyramidal neurons. This needs to be clarified.

Second, blocking the DREADD-Gq and adrenergic receptors-induced bursting pattern in PV and pyramidal neurons with YM-254890 supports the hypothesis of endogenous Gq receptors implication. However, acting on downstream effectors such as the adenylyl cyclase, or cAMP would be a stronger demonstrations.

Third, the authors mention that 'blockade of fast GABAergic inhibitory synaptic transmission with picrotoxin (50 μ M) did not affect the pattern of action potential bursts stimulated by CNO in PV interneurons (n=3, data not shown)'. As they make an important claim about this experiment, these data need to be shown. Moreover, n=3 is a very low number of cells especially if they are not all recorded from 3 different mice.

Abstract

Line 21: *chemogenetic* Gq activation

Line 24 *chemogenetic* Gq activation *also* induced

Introduction

Line 75: Gq signaling by *exogenous expression* of the Gq-coupled designer receptor hM3D, or...

Results

Line 86: *Chemogenetic* Gq activation

Line 90: viral delivery of the Gq-DREADDs: this sentence is not correct. The viral vector was delivered to *express* the Gq-DREADDs

Line 142: the authors should consider to also cite PMID: 23805227, which specifically addressed the role of perisomatic inhibition on Pyramidal neuron activity in the hippocampus.

Figures

1A: write the mouse line (PV-cre) before (on the top of) the viral construct

Fuse panel A+B in one panel A.

1D: add a schematic of the BLA and a patch pipette targeting a pyramidal neuron (as in Figure 2A)

1D: include the time scale in the colored rectangles.

1E: specify the number of mice, cells and bursts analyzed

1I: add a schematic of the BLA and 2 patch pipettes: one targeting a pyramidal neuron in whole cell, and the other a PV iNn in loose-seal (as in Figure 2A)

5B and D: indicated from how many mice the 7 PV neurons were recorded

Overall, including the number of cells/mice on the graphs would increase the impact of the Figures.

We thank the Reviewers for their careful consideration of our manuscript. They will find below our responses to their comments and suggestions. We have addressed each of their comments and believe that the paper is much improved as a result. The revised paper includes data and figures from additional experiments where indicated, further analyses of field potential oscillations, and changes in the discussion. All substantive revisions in the manuscript are highlighted with red text.

Reviewer #1 (Remarks to the Author):

The manuscript by Fu et al. identifies that activation of Gq via DREADD or alpha1A adrenoreceptors induces an intrinsic burst firing mode in BLA PV neurons, leading to synchronized inhibitory drive in BLA principal neurons, changes in the local LFP and enhanced fear retrieval in a loss-of-function mouse line. The findings are highly interesting. However, the strong conclusions in relation to the in vivo function of this cellular burst phenotype are currently somewhat weakly supported by the data. Furthermore, some controls are missing.

The authors are encouraged that the Reviewer finds the current study “highly interesting” and we appreciate the opportunity to clarify some of the methodology and interpretation of the results. This study represents the first demonstration that neuromodulators signaling through Gq can alter both network and behavioral states. Although there remains much work to be done to fully understand the role of neuromodulation-dependent transitions between network and behavioral states, these findings represent a major step forward in our current understanding.

1. Fig. 1C: The GAD-GFP label is unclear. Is this a mouse line, viral expression, antibody stain?

Thank you for bringing this lack of clarity to our attention. This has been revised in the legend to Fig. 1 to clarify that the GAD-GFP label is in a GAD67-GFP mouse line.

2. Fig. 1E, Fig. 3E,J: Given that the bursts have different lengths and IPSC numbers, wouldn't it be more intuitive to plot the Intra-burst IPSC frequency vs. the relative burst length? This will make the data more comparable across cells and events.

The authors thank the Reviewer for this suggestion. This has been changed in the relevant figures according to the reviewer's suggestion, which provides a better representation of the characteristic fast acceleration of the IPSC burst.

3. Fig. 1K: Vertical units / scale bar missing.

The vertical scale has been added.

4. Fig. 1I-K: Results are based on one example cell. Was this reproducible? If yes, summary data is missing.

This experiment was repeated in 5 PV neuron–principal neuron pairs. The PV neuron spike bursts were time locked to one of the subgroups of repetitive IPSC bursts (usually numbering 3 to 5, see Fig. 2) in all 5 of the paired recordings. Given the differences in burst frequency and burst subgroups from one pair to the next, this finding is difficult to summarize graphically. We have included the statement in the text that it was found in all 5 of the paired recordings (line 142).

5. Fig. 1L: Axis units are missing.

This has been corrected.

6. Fig. 2B: Classification of synchronized events is not clear. The authors should also provide temporally expanded representative traces to illustrate the event synchrony.

As recommended, expansions of regions of the bursts have been added in a separate panel to the figure, which better illustrates the synchronization of the intra-burst IPSCs. Thank you for this suggestion.

7. Fig. 3 I: Units / scale bars are missing.

This has been corrected.

8. Fig. 5, 6, 7: These experiments miss controls (e.g., mCherry expressing animals similar to the controls in Fig. 8). It can't be ruled out that the observed effects are mediated by CNO alone (or drift) irrespective of DREADD expression.

The authors thank the Reviewer for making us aware that the control experiments were not clear in the original text. The saline and CNO controls were previously reported in the manuscript, though in a disorganized manner. These data have been reorganized and included in Supplementary Fig. 7. These experiments include two control groups: a vehicle DREADD control and a CNO no-virus control.

9. Fig. 5: Quantification across cells should be provided.

We have now quantified spiking patterns using interspike interval (ISI), predominant frequency of ISI, and the coefficient of variation of ISI, the illustrations of which we have included in Fig. 5 in the revision.

10. Fig 6, 7: The LFP analysis window size is 5s. This seems long and the window size for the time frequency analysis should be adjusted to a specific frequency (e.g. 5-10 cycle of an oscillation to estimate theta, gamma separately). The quantification of "normalized power area" is not clear. A wavelet spectrogram would help to demonstrate the change of power in different frequency bands. Comparing the mean power of gamma oscillations in a 30 min window is problematic due to the transient nature of gamma events (alternative approaches: changes in the tail of the power distribution, bootstrapping).

We appreciate the reviewer's concerns regarding LFP analysis window size consistency for different frequency analyses and the transient nature of gamma events. While the number of gamma cycles outnumbers theta cycles in a 5 second window, we believe the greater number of gamma cycles per window results in a more accurate estimation of gamma power per window, rather than a diminished estimation ability, given we capture at least roughly 10 cycles per window. However, to demonstrate the maintenance of our observed effects with a more consistent time window, we reran our FFT algorithm on our in vivo Gq-DREADD data using a 0.25 second analysis window and included the results here (Response Fig. 1). A 0.25 second FFT analysis window should include 10 cycles of a 40 Hz oscillation (the low end of our gamma band), consistent with the 10 cycles of a 2 Hz oscillation (the low end of our theta band) captured with a 5 second window. Additionally, we have included spectrograms using our 5 second FFT data in all the in vivo figures of the manuscript, as suggested by the reviewer to

improve accessibility. This analysis produced similar results to those reported in the manuscript, indicating that the size of the analysis window we used did not bias the results in this dataset.

Response Figure 1. 0.25 second (10 cycle) FFT analysis window produces similar results to the 5 second window analysis. (A) Spectrogram illustrating normalized power across frequencies over the last 30 minutes of saline and CNO treatments in PV hM3D-expressing animals. Note that less than 10 cycles will be captured for frequencies below 40 Hz with this 0.25 second window size. (B) Mean (+/- SEM) normalized power across treatments (saline vs CNO) and frequency bands. Values from last 30 minutes of recording were used for analysis, illustrated by the highlighted segments in C. Two-way ANOVA [treatment x frequency], $F(1.058, 9.523) = 6.970$, $p = 0.0246$; Sidak's multiple comparisons test: 2-6 Hz, $p = 0.2857$; 6-12 Hz, $p = 0.3196$; 15-30 Hz, $p = 0.6891$; 40-70 Hz, $p = 0.0326$; 70-120 Hz, $p = 0.0135$. $n = 10$. *, $p < 0.05$. ns = not significant. (C) Normalized power across time for 40-70Hz (top) and 70-120 Hz (bottom). Values averaged in 1-min bins. Colored lines = average power over time, colored shaded areas = SEM. Highlighted region indicates last 30 minutes included in analysis. (D) Average probability density plots illustrating the distribution of gamma powers across treatments for 40-70 Hz (top) and 70-120 Hz (bottom). Colored solid lines = average probability, colored shaded areas = SEM. Vertical lines indicate average threshold value for distribution tails, where threshold = 1 standard deviation above the mean.

The Reviewer provides an excellent suggestion for analyzing the shifts in gamma power in this dataset, which we have now adopted for the entire paper. We investigated changes in the tails of the gamma power distributions across our *in vivo* experiments (but not slice experiments as our *ex vivo* model displays continuous gamma cycling (Supplemental Fig. 5)) and included our results in the corresponding figures throughout the manuscript. In brief, analysis of gamma distribution tails (bouts of gamma activity with powers at least 1 standard deviation above the mean for each treatment) supported our findings that PV Gq-DREADD and BLA $\alpha 1A$ receptor activation suppresses local gamma oscillatory power *in vivo* and provided a convincing demonstration of changes in gamma power associated with Gq signaling in the BLA. Note that with an increased number of replicates in *in vivo* field potential recordings and the further analysis of the LFPs, the significance in the changes in theta waves with Gq-DREADD activation in PV neurons was lost (see Fig. 6E) and the change in slow gamma oscillations with $\alpha 1A$ receptor activation (see Fig. 7D-F) was strengthened.

11. Fig 6, 7: Theta events are state dependent. Did the analysis of theta power account for the behavioral state (eg. similar running speeds, active vs. inactive periods). Drug-related changes in behavioural state might confound the LFP analysis and should be ruled out.

The Reviewer is correct that theta events have previously been reported to be state dependent, such as when the animal is engaged in locomotor activity, especially in other regions such as the hippocampus. To address the concern that focused intra-BLA norepinephrine infusion may be altering locomotor activity and, thereby, altering theta

oscillations, we analyzed mouse electromyographic (EMG) activity from our intra-BLA infusion experiments and included the results here (Response Fig. 2). We did not observe an effect of NE infusion into the BLA on locomotor activity. We provide the results here rather than in the text in order not to distract from our larger point that our findings demonstrate the alteration of network/behavioral states by Gq neuromodulation of BLA PV interneurons. However, if the Reviewer insists that these data be included in the manuscript, we can add them to the Supplemental Information. As stated above, with the addition of replicates to the data set for the revision, the statistical significance of the increase in theta power by Gq activation was lost, although the trend remains. It is important to note that the Reviewer's point still holds, and our labs are pursuing the relationship between neuromodulation of network states and the relationship to behavioral states.

Response Figure 2. Intra-BLA norepinephrine does not influence EMG activity. Percent time moving per treatment following saline 2 and norepinephrine (NE) infusions, extrapolated from EMG root-mean-square (RMS) values, across several event detection thresholds. Threshold = (one standard deviation * detection threshold) above the 5th percentile of EMG RMS values. Paired *t* tests, *p* = 0.7655, 0.8820, 0.9815, 0.8976 for detection thresholds 0.8, 1.0, 1.2, 1.4, respectively. *n* = 8. *ns* = not significant.

12. Fig. 8 / page 23. l. 348-354: The behavioural data does not seem supportive of the proposed conclusion, i.e., that neuromodulation induced burst firing induces 'transition between ... behavioural states. The authors speculate that the loss of fear retrieval is due to the PV-cre line, but not the general loss of alpha1A receptors. However, the PV-cre line has been successfully used in fear learning paradigms before and rescuing fear retrieval in this learning-deficient intercross line does not provide a convincing mechanistic link of transitions of behavioural states and burst firing in BLA PV neurons. Demonstrating that CNO induced burst firing in PV-neurons is correlated with enhanced fear retrieval would be an alternative, more convincing experiment. In addition, the data that the alpha1A KO mouse has no learning deficit should be shown.

We agree with the reviewer that the loss of fear retrieval in the control PV-cre x adra1A KO mouse is unexpected. As stated, we found no effect of global adra1KO alone on fear acquisition or retrieval compared to the wild-type mouse, which is now shown in Supplemental Fig. 9. We have also tested control virus injection into the BLA of CCK-cre x adra1A KO mice, and those mice did not show a deficit in fear retrieval (we prefer not to include those data here because they are part of a parallel study, although we will if requested). As raised by the

Reviewer, others have tested the PV-cre mouse and not seen a similar deficit in fear retrieval. Together, these data suggest that the deficit in fear retrieval is not caused by the PV-cre line per se, but rather by the adra1A KO x PV-cre cross. We can only speculate, therefore, that Cre expression in adra1A-lacking PV neurons throughout the brain suppresses fear retrieval. Nonetheless, the facilitation of fear retrieval with targeted rescue of the adra1A receptor and with hM3D expression selectively in BLA PV neurons clearly demonstrates that Gq activation in PV neurons controls changes in fear behavioral state.

We agree with the Reviewer that it is an overstatement to state that our data demonstrate that neuromodulation-induced burst firing induces “transition between... behavioral states”. We have toned down the language in the manuscript alluding to this direct relationship. We also agree that it would be highly valuable to demonstrate a correlation between the Gq-induced transition to burst firing in PV neurons with the transition in behavioral state. However, we respectfully argue that the experiments necessary to demonstrate this correlation directly, because of the spatial distribution of the PV neurons in the BLA, would require in vivo unit recordings of photo-tagged PV neurons or mini-scope calcium imaging of individual GCAMP-transduced PV neurons, either of which would require 6-12 months to complete, if successful, because of the sparsity of the PV neurons, the technical difficulty of the experiments, and the fact that we do not currently have these techniques in hand. Respectfully, we would argue that the transition of PV neurons to burst firing in vitro and facilitation of fear retrieval in vivo by the same targeted activation of Gq-coupled receptors selectively in PV neurons does in fact show indirectly a convincing correlation of BLA PV neuron burst firing with changes in behavioral state, which we hope to be able to confirm with a more direct demonstration in the future. However, we feel that the novel findings in the current study, demonstrating a role for neuromodulation in altering network activity in the BLA associated with altered behavioral states, is a major contribution to the field and furthers our knowledge of how neuromodulation impacts network states.

13. Fig. 8H: The timeline suggests that only DREADD animals were injected with CNO. However, the mCherry animals should have been injected as well, otherwise they are not controlling for potential CNO effects. This should be clarified or the appropriate controls should be added.

We apologize that this control group was not clear in the original manuscript. The mCherry controls were tested for CNO effects, which was represented by the blue symbols in Fig. 8J, but was not made clear in the timeline or in the description of the experiments in the text. These control virus experiments have now been better described in the text in line 380 and have been included with hM3D in the CNO injections indicated in the timeline in Fig. 8H (hM3D/ctrl).

14. It is not clear how general Gq activation leads to intrinsic burst firing in PV neurons. Potential mechanisms should be discussed.

We agree that the cellular mechanism of Gq activation of intrinsic repetitive burst firing in PV neurons is interesting. We have begun to test for mechanisms but have not identified any yet (see response below). We have added to the Discussion possible mechanisms that we are targeting in lines 386-389.

Reviewer #2 (Remarks to the Author):

In this study the authors examine the impact of Gq signaling in the basolateral amygdala. Using a variety of techniques and transgenic lines they show that activation of Gq receptors in Parvalbumin-expressing interneurons induces a stereotypical pattern of bursting electrophysiological activity. This results in patterns of bursting inhibitory input to glutamatergic principal neurons, converting activity of these neurons to a bursting oscillatory pattern. This cellular activity is seen in the local field potential as a change in gamma band oscillations. Also using transgenic mouse lines and pharmacology they show that in the BLA this modulation can be mediated by activation of alpha1A adrenoceptor receptors. Finally, they show that this Gq activity modulates the expression of conditioned fear. The experiments in this paper are complex and provide a large body of data that convincingly backs the above conclusions. I congratulate the authors on their very well thought out and complex experiments. The data shown is clear and the conclusions they reach well supported by the data.

The authors are appreciative that the Reviewer recognizes the complexity of our study and finds the experiments well thought out and the conclusions supported by the data.

That said, I have several issues that should be addressed. Firstly, in almost all the experiments shown Gq receptors are activated by bath application of CNO or norepinephrine coupled with expression of different constructs in PV-interneurons (using a PV-cre line crossed with other transgenic lines). The mechanism of how the discharge pattern of these interneurons changes is not addressed but this is not the point of the paper.

The Reviewer highlights an interesting, yet unresolved aspect of the current study. The ability of Gq signaling (NE or DREADDs) to dramatically alter the firing pattern of PV interneurons was striking and inspired us to examine the impact on network activity coordinated by this subset of interneurons. The current study focuses on how this mechanism impacts network states and associated behavioral states; however, we agree that the mechanisms mediating the changes in PV interneuron activity also present an interesting question, which will be the focus of future studies, but outside the scope of the current study. We have begun to address this pharmacologically but have not yet been able to identify a specific mechanism. We have added a statement to this effect to the Discussion (lines 386-389)

Note that the impact of norepinephrine input to the BLA on fear expression has been shown by Uematsu et al (Nature Neuroscience 20, 1602–1611 2017) and the data here (Figure 8) reproduces their findings. Surprisingly this paper is not cited.

This was an oversight that has been corrected (line 438). Thank you for pointing it out.

This method activates either adrenergic receptors on all PV-interneurons or activates drives Gq widely. Perhaps as expected, bath application of serotonin, also coupled to Gq receptors can reproduce the effects of norepinephrine. However, in vivo, the receptors are driven by transmitter release from ascending noradrenergic systems. It is not at all clear what the impact of this input would have on network activity - where there is some diffuse transmission, much of the impact of these transmitters is synaptic over short periods of release. It is not clear if any of the network activity would be seen when these systems are active. If this input to the BLA also changes PV cell activity and the activity of principal neurons is not known. While the behavioural experiments do go some way to addressing this point, it would have been very useful to test the impact of synaptically released transmitter. One option is to use a TH-cre line and optogenetics to directly release NE into the BLA. While this is not a major issue with the data shown in this study - I feel that it should be addressed in the discussion.

The Reviewer points out that these studies rely on either artificial Gq signaling (DREADDs) or exogenous administration of NE, which should be made clear in the manuscript. We now specify the nature of the approach in the section subheadings and highlight the need for future experiments on endogenous neuromodulator release, in the Discussion on p. 26 (lines 388-397). These findings take the first steps in improving our understanding of the impact of neuromodulation on BLA network states and related behavioral states; our future studies will build on these foundational experiments to explore the impact of endogenous neuromodulators and the context under which they are engaged.

Secondly, all manipulations have been done using PV-cre mice. There is no doubt that PV cells are the main basket interneurons in the BLA. However, there is another class of axo-axonic interneurons in the BLA that also innervate many hundreds of principal cells. The procedures used here do not differentiate these two types. However, the manuscript is written entirely from the perspective of PV-basket cells - this separation needs to be made clear and some evidence provided for the role of basket interneurons. They do address the issue of CCK interneurons and allude to SOM-interneurons but not the axo-axonic cells. This is an important distinction.

We appreciate that the Reviewer recognizes our discussion of the potential influence of other interneuron subtypes (CCK and SOM) on these outcome measures. In response to the Reviewer's suggestion, we have also added a discussion of chandelier cells, which may also contribute to the PV neuron response we describe, since they also express Cre in the PV-Cre mouse line. We have included chandelier cells in the discussion of the tight regulation of BLA principal neuron output by basket cells (lines 407-409).

Minor:

For the LFP data - it would be good show some straightforward power-spectra for the non-aficionado.

This is an excellent suggestion that not only highlights the findings of the manuscript but also makes the data more visually compelling. We have provided spectrograms to illustrate power-spectra data across frequency bands over time in addition to the normalized power area plots for frequency bands with significant effects.

I was very surprised by the very high concentration of compounds used - norepinephrine up to 100 μM while the EC50 for these compounds acting at G-protein coupled receptors is in the low micromolar range. While the pharmacological assessment is fine - there is always a change of off-target actions with these high concentrations.

We agree that 100 μM is a high concentration. However, we performed a concentration-response analysis and found that the bursting activity begins to emerge at a concentration of 20 μM , albeit less consistently and robustly than at higher concentrations (see Supplementary Fig. 2). Based on this empirical evidence, we decided to use the higher concentration to obtain a consistent response and to account for diffusion in the microinfusion experiments. This concentration falls within the concentration range used in brain slice studies, which are often higher than in cell cultures where EC50's are often calculated. While we can speculate as to the local concentration of NE in or around release sites, which could reach millimolar concentrations if similar to glutamate and GABA synapses, we cannot say with certainty that what we are reporting is physiological. Nevertheless, the highly reproducible and stereotypic pattern of the robust bursting activity brought out by the NE as well as the Gq-DREADD reveals a distinct intrinsic bursting mechanism in PV cells that is physiological, and the similar activity induced by 5-HT and the Gq-DREADD suggests that the Gq-induced bursting activity is part of the PV

neuron's response range. We have discussed this in the Discussion (lines 390-399).

Reviewer #3 (Remarks to the Author):

In this study Fu et al. present very interesting findings on the electrophysiological and behavioral impact of PV interneurons activation through GPCR Gq protein activation (exogenous DREADDs and endogenous adrenergic receptors) in the basolateral amygdala (BLA). The results provide a potential mechanism for the role of the adrenergic-2A receptor in fear encoding in the BLA, and might have a strong impact on the field of learning and memory. However, I have concerns regarding the description and interpretation of the DREADDs experiments, and other points.

The authors thank the Reviewer for her/his thoughtful review of the manuscript and encouraging words.

First, the authors perform manipulation of exogenous Gq receptors (DREADDs), which is very different than increasing the activity of endogenous Gq receptors. The whole manuscript needs to make this point very clear, and the results should be interpreted as such. Indeed, the DREADD activation does not demonstrate that endogenous Gq activation in PV interneurons induces bursting of these neurons, as well as IPSC bursts in pyramidal neurons. The DREADDs experiments only suggests that endogenous Gq activation induces bursting in BLA PV interneurons and pyramidal neurons. This needs to be clarified.

The Reviewer highlights an important detail regarding the reliance on either artificial Gq signaling (DREADDs) or exogenous administration of NE, which was not made clear in the manuscript. We now specify the nature of the approach in the Abstract and the section subheadings, and highlight the caveats of these approaches in the discussion (lines 390-399). We have added a statement to the discussion distinguishing between exogenous hM3D activation and endogenous $\alpha 1A$ receptor activation, lines 398-399, and have implemented all the specific changes suggested by the Reviewer below.

Second, blocking the DREADD-Gq and adrenergic receptors-induced bursting pattern in PV and pyramidal neurons with YM-254890 supports the hypothesis of endogenous Gq receptors implication. However, acting on downstream effectors such as the adenylyl cyclase, or cAMP would be a stronger demonstration.

The Reviewer is correct that while our data, including Gq-DREADD activation, Gq-coupled $\alpha 1A$ adrenoreceptor activation and inhibition, and Gq subunit inhibition, convincingly demonstrate the Gq dependence of the bursting activity induced in PV interneurons, they fail to pin down the signal transduction mechanisms downstream from Gq. While this is an important direction of research, the current study focuses on how this mechanism impacts network states and associated behavioral states, at the expense of postsynaptic signal transduction mechanisms. We agree that additional information regarding the downstream mediators is important, and we have begun to test several downstream mediators in the Gq signaling pathway, including PLC and PKC activities and Ca release from intracellular stores, but to date we have not obtained compelling findings regarding the intracellular signaling cascades. The Gq signaling that leads to the induction of intrinsic bursting in the PV neurons appears, therefore, to tap a non-canonical signaling mechanism, the characterization of which is proving complex and, we would argue, more involved than what would fall within the scope of this study. We hope to be able to solve this riddle in future studies

Third, the authors mention that ‘blockade of fast GABAergic inhibitory synaptic transmission with picrotoxin (50 μ M) did not affect the pattern of action potential bursts stimulated by CNO in PV interneurons (n=3, data not shown)’. As they make an important claim about this experiment, these data need to be shown. Moreover, n=3 is a very low number of cells especially if they are not all recorded from 3 different mice.

The original data were from 3 cells recorded in slices from the same mouse. We have now added experiments in an additional 2 cells from a second mouse. While this is not a large “n”, blocking GABA_A receptor and ionotropic glutamate receptor signaling failed to suppress the PV neuron bursting in all 5 cells tested, such that the result is quite robust. These data have been added to the manuscript text (lines 125-126) and included in Supplemental Fig. 1G.

Abstract

Line 21: *chemogenetic* Gq activation

Line 24 *chemogenetic* Gq activation *also* induced

Suggested changes made.

Introduction

Line 75: Gq signaling by *exogenous expression* of the Gq-coupled designer receptor hM3D, or...

Suggested changes made.

Results

Line 86: *Chemogenetic* Gq activation

Line 90: viral delivery of the Gq-DREADDs: this sentence is not correct. The viral vector was delivered to *express* the Gq-DREADDs

This sentence has been reworded to make this distinction.

Line 142: the authors should consider to also cite PMID: 23805227, which specifically addressed the role of perisomatic inhibition on Pyramidal neuron activity in the hippocampus.

We have added the suggested reference.

Figures

1A: write the mouse line (PV-cre) before (on the top of) the viral construct

Fuse panel A+B in one panel A.

The suggested changes have been made to Fig. 1.

1D: add a schematic of the BLA and a patch pipette targeting a pyramidal neuron (as in Figure 2A)

The suggested schematic has been added to Fig. 1.

1D: include the time scale in the colored rectangles.

This change has been made.

1E: specify the number of mice, cells and bursts analyzed

This information has been added to the figure legends, where missing, and is highlighted by red text.

1I: add a schematic of the BLA and 2 patch pipettes: one targeting a pyramidal neuron in whole cell, and the other a PV iNn in loose-seal (as in Figure 2A)

The suggested schematic has been added to Fig. 1.

5B and D: indicated from how many mice the 7 PV neurons were recorded

This has been added.

Overall, including the number of cells/mice on the graphs would increase the impact of the Figures.

This information has been added to all the figure legends.

REVIEWERS' COMMENTS

Reviewer #1 (Remarks to the Author):

The manuscript is significantly approved. I have few comments / suggestions left:

The visual contrast of the new spectrograms (e.g. Fig. 6D) is relatively low in the frequency ranges with the strongest effect (analysis in 6E-F). This could potentially be improved by whitening of the signal. Furthermore, it might be useful for the reader if the authors included Response Figure 1 (different analysis window) as a supplementary figure.

Reviewer #2 (Remarks to the Author):

The authors have addressed most of the concerns that I raised. With respect to the different types of PV interneurons. They now address the fact that there are different classes of PNV interneurons. However, what is not said is the following. It is not known what classes of noradrenergic receptors are expressed by different interneurons nor what the impact of NE has on these different classes. In particular whether they all respond to synaptically released NE. In the discussion - it should be made explicit that how these network effects are mediated is not clear.

Reviewer #3 (Remarks to the Author):

The revised version of the manuscript has greatly improved, including additional data, and clarification of multiple concepts. The authors have addressed all my comments, and only one remains to be fully addressed. Indeed, although the authors clarified the difference between exogenous (chemogenetic) and endogenous (adrenergic-1A) Gq activation, the title and abstract still need to include this concept. Moreover the title is overstating the results as in the behavioral experiments the authors did not record the bursting activity. Thus the title needs to be adjusted.

I suggest one of the following titles [1] "Activation of endogenous and exogenous excitatory GPCR in parvalbumin interneurons of the basolateral amygdala induces burst firing and facilitates fear memory retrieval"

Or [2] "Adrenergic-1A and chemogenetic Gq receptors in parvalbumin interneurons of the basolateral amygdala induce burst firing and facilitate fear memory retrieval"

Similarly the abstract should be more specific as it is hardly understandable without reading the full manuscript.

Lin 25-26: "Chemogenetic Gq activation also induced a transition from tonic to phasic firing in parvalbumin 25 neurons and principal neurons, which" is redundant with the 2 previous sentences – please replace with "interestingly, these currents and firing pattern suppressed gamma oscillations in vivo and in ex vivo slices."

Line 27-28: Consider replacing with: "Finally, endogenous 1A and exogenous DREADD Gq activation in parvalbumin interneurons facilitated fear memory recall, which is consistent with BLA gamma suppression previously reported during conditioned fear expression."

Line 28-30: The last sentence is unclear partly due to the lack of definition of what the authors refer to with "a fear-associated network" and "behavioral state". This sentence should be reworded to be more specific and describing the results of the study.

Minor comments

Figure 2 C: please make 2 separate graphs

Figure 6 title: consider replacing 'network' with 'oscillatory'

Reviewer #1 (Remarks to the Author):

The manuscript is significantly approved. I have few comments / suggestions left:

The visual contrast of the new spectrograms (e.g. Fig. 6D) is relatively low in the frequency ranges with the strongest effect (analysis in 6E-F). This could potentially be improved by whitening of the signal.

We have included updated spectrograms to improve visual contrast. As suggested, we whitened the signal of all spectrograms by 40%.

Furthermore, it might be useful for the reader if the authors included Response Figure 1 (different analysis window) as a supplementary figure.

We have added the 0.25 second analysis window figure to the manuscript as Supplementary Figure 8 and included the following description in the Methods section (lines 668-670) "Importantly, an additional analysis using a 0.25-second window, which captured 10 cycles of the slowest gamma frequency (40 Hz), produced results similar to the 5-second window analysis (which captured 10 cycles of the slowest theta frequency (2 Hz)) (Supplementary Fig. 8).

Reviewer #2 (Remarks to the Author):

The authors have addressed most of the concerns that I raised. With respect to the different types of PV interneurons. They now address the fact that there are different classes of PV interneurons. However, what is not said is the following. It is not known what classes of noradrenergic receptors are expressed by different interneurons nor what the impact of NE has on these different classes. In particular whether they all respond to synaptically released NE. In the discussion - it should be made explicit that how these network effects are mediated is not clear.

We have added the following paragraph to the Discussion to address this concern (lines 408-416): "Indeed, we found that inhibiting calcium-dependent release from PV and CCK neuron axons blocked ~50% of the NE-induced increase in sIPSC frequency (Supplementary Fig. 3), suggesting that the remaining ~50% of the NE response was generated by one or more other interneuron subtypes, such as the somatostatin-expressing GABA neurons. Interestingly, the response was blocked completely by an $\alpha 1A$ receptor antagonist (Supplementary Fig. 3), was insensitive to β adrenoreceptor blockade (Supplementary Fig. 3), and was lost entirely in the $\alpha 1A$ receptor knockout mouse (Supplementary Fig. 9), which indicated that the NE stimulation of GABA neurons in the BLA is mediated exclusively by $\alpha 1A$ receptors. The relative effects of synaptically released NE on different inhibitory interneuronal circuits is not known and requires further study."

Reviewer #3 (Remarks to the Author):

The revised version of the manuscript has greatly improved, including additional data, and clarification of multiple concepts. The authors have addressed all my comments, and only one remains to be fully addressed. Indeed, although the authors clarified the difference between

exogenous (chemogenetic) and endogenous (adrenergic-1A) Gq activation, the title and abstract still need to include this concept.

Moreover the title is overstating the results as in the behavioral experiments the authors did not record the bursting activity. Thus the title needs to be adjusted.

I suggest one of the following titles [1] "Activation of endogenous and exogenous excitatory GPCR in parvalbumin interneurons of the basolateral amygdala induces burst firing and facilitates fear memory retrieval"

Or [2] "Adrenergic-1A and chemogenetic Gq receptors in parvalbumin interneurons of the basolateral amygdala induce burst firing and facilitate fear memory retrieval"

The reviewer raises a valid concern. We have changed the title to reflect the lack of direct demonstration of the Gq-induced behavioral changes being driven by the Gq-induced bursting behavior in PV interneurons. We preferred to change the title to "Gq neuromodulation of BLA parvalbumin interneurons induces burst firing and mediates fear-associated network and behavioral state transition in mice" from the old title of "Neuromodulation-induced burst firing in BLA parvalbumin interneurons mediates transition between fear-associated network and behavioral states", which dissociates the Gq-induced bursting activity in the PV neurons from the PV neuron Gq-induced network and behavioral state changes.

Similarly the abstract should be more specific as it is hardly understandable without reading the full manuscript.

Lin 25-26: "Chemogenetic Gq activation also induced a transition from tonic to phasic firing in parvalbumin 25 neurons and principal neurons, which" is redundant with the 2 previous sentences – please replace with "interestingly, these currents and firing pattern suppressed gamma oscillations in vivo and in ex vivo slices."

We have rewritten the abstract as suggested in an attempt to make the abstract more understandable.

Line 27-28: Consider replacing with: "Finally, endogenous 1A and exogenous DREADD Gq activation in parvalbumin interneurons facilitated fear memory recall, which is consistent with BLA gamma suppression previously reported during conditioned fear expression."

Line 28-30: The last sentence is unclear partly due to the lack of definition of what the authors refer to with "a fear-associated network" and "behavioral state". This sentence should be reworded to be more specific and describing the results of the study.

We have changed the concluding sentence to the following: "Thus, here we identify a neuromodulatory mechanism in PV inhibitory interneurons of the BLA which regulates BLA network oscillations and fear memory recall."

Minor comments

Figure 2 C: please make 2 separate graphs

In order to avoid including a single-column bar graph, we have dropped the synchronized burst ratio bar from the bar graph in Fig. 2C and report it in the text (lines 141-144), and illustrate only the number of synchronized bursts in the graph.

Figure 6 title: consider replacing 'network' with 'oscillatory'

This change has been made